# Generalized Fisher-Weighted SVD: Scalable Kronecker-Factored Fisher Approximation for Compressing Large Language Models

## Abstract

The Fisher information is a fundamental concept for characterizing the sensitivity of parameters in neural networks. However, leveraging the full observed Fisher information is too expensive for large models, so most methods rely on simple diagonal approximations. While efficient, this approach ignores parameter correlations, often resulting in reduced performance on downstream tasks. In this work, we mitigate these limitations and propose Generalized Fisher-Weighted SVD (GFWSVD) — a fully deterministic post-training LLM compression technique that accounts for both diagonal and off-diagonal elements of the Fisher information matrix, providing a more accurate reflection of parameter importance. To make the method tractable, we introduce a scalable adaptation of the Kronecker-factored approximation algorithm for the observed Fisher information. We demonstrate the effectiveness of our method on LLM compression, showing improvements over existing compression baselines.

## 1 Introduction

The Fisher Information Matrix (FIM) (Fisher, 1992) is widely employed in neural networks to enhance the efficiency of models, particularly in the context of training and inference. However, computing and leveraging the full Fisher information is computationally prohibitive for deep networks. To make the problem tractable, existing methods adopt simplified approximations – most commonly, assuming that the Fisher matrix is diagonal (Wu et al., 2024; Frankle & Carbin, 2019; Soen & Sun, 2024). While efficient, this assumption discards valuable information about parameter correlations.

One key application of FIM is low-rank compression of large language models (LLMs). However, the standard low-rank approach — Singular Value Decomposition (SVD) — often leads to suboptimal performance. To mitigate this, weighted SVD methods aim to align the optimization objective with the target task (Yuan et al., 2023; Hsu et al., 2022). Fisher-Weighted SVD (FWSVD) (Hsu et al., 2022) uses Fisher information to assign importance to parameters. However, FWSVD utilizes only the diagonal part of FIM and treats each row as independent, which can lead to poor retention of task-critical components.

In contrast, we propose a more accurate weighted SVD method: **Generalized Fisher-Weighted SVD (GFWSVD)**. Our approach leverages a Kronecker factorization of the full FIM to derive two sensitivity matrices, which are integrated into a generalized SVD framework. To overcome the high computational cost of factorizing the full Fisher matrix, we introduce a scalable adaptation of the Kronecker decomposition algorithm. We compare our method with various low-rank compression approaches for large models — those using Fisher information (Fisher-Weighted SVD), and those leveraging activation statistics (ASVD (Yuan et al., 2023), SVD-LLM (Wang et al., 2025c)) — and observe consistent improvements in downstream task performance.

To summarize, our main contributions are as follows:

- We introduce **Generalized Fisher-Weighted SVD (GFWSVD)**, a new weighted SVD-based fully deterministic method for compressing large language models, which leverages the Kronecker-decomposed Fisher information that encodes both row-wise and column-wise

parameter correlations. We prove that **GFWSVD** is a generalization of FWSVD (Hsu et al., 2022).

- We propose a computationally effective adaptation of the Kronecker decomposition algorithm for the Fisher information matrix (FIM) that captures its full structure without relying on diagonal or other simplifying approximations.

- We empirically show that our method preserves model performance under compression while maintaining efficiency, outperforming existing techniques within its class.

## 2 RELATED WORK

Fisher information is a fundamental tool for measuring parameter importance in neural networks. It has been used to prevent catastrophic forgetting in continual learning (Kirkpatrick et al., 2017), to guide local updates in federated learning (Jhunjhunwala et al., 2024), and more recently to merge fine-tuned models at the parameter level (Lee et al., 2025). As computing the full FIM is expensive, many methods rely on structural assumptions to make it tractable. A widely used strategy is Kronecker-product factorization, which breaks the FIM into manageable components. KFAC (Grosse & Martens, 2016) introduced this idea for convolutional layers, showing that structured approximations can preserve key curvature information while cutting costs. Later work (Tang et al., 2021) improved efficiency with faster Kronecker-factored updates, while KPSVD (Koroko et al., 2023) applied singular-value constraints to enable memory-efficient FIM approximations in large models. We use a Kronecker-factored FIM approximation for model compression based on the structural approximation of every separate layer.

Post-training compression based on structural approximation has shown promising results. Such methods typically rely on a weighted decomposition of a model layer's weights, incorporating either loss-aware or activation-aware information. For instance, SparseGPT (Frantar & Alistarh, 2023) ranks weights using curvature estimates for pruning, FWSVD (Hsu et al., 2022) applies diagonal FIM approximations to guide task-aware SVD compression. As we later demonstrate, FWSVD emerges as *a special case of our more general framework*, underscoring the flexibility of our approach. AdaSVD (Li et al., 2025) distributes compression strength across layers via adaptive compensation, while ASVD (Yuan et al., 2023), NSVD (Lu et al., 2025), and SVD-LLM variants (Wang et al., 2025c;b) use activation statistics to refine truncation. Notably, many of these methods assume independent parameter contributions, which can limit task sensitivity. In contrast, our Kronecker-factored *approximation of the full observed FIM captures both row- and column-wise dependencies within weight matrices*, yielding more accurate importance estimates. There are also approaches that account for dependencies between layers in the model, rather than just correlations between parameters within a single layer: (Wang et al., 2024) uses a shared set of basis vectors to represent the weight matrices of different layers, which effectively eliminates cross-layer redundancy.

Other classes of compression methods, such as quantization, also leverage Fisher information in their setups. For instance, the YAQA (Tseng et al., 2025) quantization method extends QTIP (Tseng et al., 2024) by incorporating curvature information from the loss landscape.

In more advanced structural approximation pipelines, compression settings are optimized through training, and structural approximations are combined with additional techniques to enable more aggressive compression. For example, BLAST (Lee et al., 2024) relies on a gradient-descent–based factorization algorithm, while Dobi-SVD (Wang et al., 2025a) learns optimal singular values and then applies quantization.

## 3 BACKGROUND AND PROBLEM FORMULATION

In this section, we establish the connection between Fisher information over matrix variables drawn from Matrix-Variate Normal (MVN) distribution and our approach to approximating the Fisher matrix via a Kronecker product decomposition. We then leverage this decomposition to develop an improved compression algorithm based on the generalized SVD formulation.

### 3.1 LAYER COMPRESSION AND HESSIAN APPROXIMATION

Consider post-training weight compression as a perturbation of a model parameters $\theta \in \mathbb{R}^d$. The perturbation affects the deviation of the model's loss function $\mathcal{L}(\theta)$ in the proximity of an optimal point $\theta^\star$. Sensitivity to such perturbation can be naturally captured by the second-order expansion of the loss determined by the quadratic term involving the Hessian $H = H(\theta^\star)$ of the problem:

$$\nabla\mathcal{L} = \mathcal{L}(\theta) - \mathcal{L}(\theta^*) \approx \frac{1}{2}(\theta - \theta^\star)^\top H(\theta - \theta^\star) \tag{1}$$

Compression optimization thus corresponds to minimizing the deviation $\nabla\mathcal{L}$ with respect to a compression $\theta = \mathcal{C}(\theta^\star)$ while considering the structured curvature encoded in $H$:

$$\min_{\mathcal{C}} \ (\theta^\star - \mathcal{C}(\theta^\star))^\top H(\theta^\star - \mathcal{C}(\theta^\star)), \tag{2}$$

where the optimization task is considered over a functional family of compression methods $\mathcal{C}$.

In real-world settings, working directly with $H$ is often intractable due to its size and complex structure. Hence, solving the task in Eq. 2 also requires finding good enough approximations of $H$ that ideally capture the most important properties of the Hessian. As we show next, there is a certain class of approximations that align particularly well with our task.

### 3.2 MATRIX-VARIATE NORMAL DISTRIBUTION AND FISHER INFORMATION

The MVN distribution (Gupta & Nagar, 2018) extends the classical multivariate normal distribution to matrix-valued random variables, providing a structured approach to modeling dependencies within rows and columns. Formally, a matrix $\mathbf{X} \in \mathbb{R}^{n \times m}$ follows an MVN distribution if its entries exhibit Gaussian properties with covariance structured across both dimensions. The distribution is defined as

$$\mathbf{X} \sim \mathcal{MN}(\mathbf{M}, \mathbf{\Sigma}_1, \mathbf{\Sigma}_2), \tag{3}$$

where $\mathbf{M}$ is the mean matrix, and the (non-degenerate) covariance is expressed as a Kronecker product $\mathbf{\Sigma}_2 \otimes \mathbf{\Sigma}_1$. Here, $\mathbf{\Sigma}_1$ captures dependencies between rows, while $\mathbf{\Sigma}_2$ encodes dependencies across columns. This structure ensures that each row and column follows a well-defined correlated Gaussian distribution.

A crucial property of MVN is that its likelihood function inherently incorporates the inverse Kronecker-factored covariance, *leading to an efficient representation of second-order dependencies*. The log-probability density function of $\mathbf{X}$ has the form:

$$\log(p(\mathbf{X})) \propto -\frac{1}{2}\left(\operatorname{vec}(\mathbf{X} - \mathbf{M})^\top (\mathbf{\Sigma}_2 \otimes \mathbf{\Sigma}_1)^{-1} \operatorname{vec}(\mathbf{X} - \mathbf{M})\right) =$$

$$= -\frac{1}{2}\operatorname{tr}\left(\mathbf{\Sigma}_1^{-1}(\mathbf{X} - \mathbf{M})\mathbf{\Sigma}_2^{-1}(\mathbf{X} - \mathbf{M})^\top\right) \tag{4}$$

Maximization of log-likelihood leads to minimization of trace in Eq. 4, which yields the Generalized Least Squares Matrix Decomposition problem[1] (Allen et al., 2014):

$$\min_{\operatorname{rank}(\mathbf{X}) \leq r} \left\|\mathbf{\Sigma}_1^{-\frac{1}{2}}(\mathbf{X} - \mathbf{M})\mathbf{\Sigma}_2^{-\frac{1}{2}}\right\|_F^2, \tag{5}$$

directly connected to the Generalized Singular Value Decomposition (GSVD) (Golub & Van Loan, 2013). This problem can be straightforwardly solved by means of standard SVD (Abdi, 2006):

$$\mathbf{X} = \mathbf{\Sigma}_1^{\frac{1}{2}}\hat{\mathbf{U}}\hat{\mathbf{S}}\hat{\mathbf{V}}^\top\mathbf{\Sigma}_2^{\frac{1}{2}} \tag{6}$$

where $\hat{\mathbf{U}}\hat{\mathbf{S}}\hat{\mathbf{V}}^\top = \operatorname{SVD}_r(\mathbf{\Sigma}_1^{-\frac{1}{2}}\mathbf{M}\mathbf{\Sigma}_2^{-\frac{1}{2}})$. We note that the result also holds in the case when matrix square roots are replaced with the corresponding Cholesky factors, which are typically easier to find.

Under regular conditions (e.g., smooth differentiability and proper statistical properties), Fisher Information $\mathcal{I}_F$ serves as an expectation of the local curvature (second derivative) of the likelihood function. Importantly, by taking derivatives of the MVN likelihood function with respect to $\mathbf{M}$, it is easy to show that the corresponding *Hessian directly coincides with Fisher Information at the MLE solution*, e.i., $\mathcal{I}_F = H(\mathbf{M}) = \mathbf{\Sigma}_2^{-1} \otimes \mathbf{\Sigma}_1^{-1}$. This formulation provides a natural bridge between the selection of an optimal compression algorithm $\mathcal{C}$ from Eq. 2 and Fisher Information, which we establish next.

---

[1] Following the notation of Allen et al. (2014), we define $\mathbf{A} = \mathbf{A}^{\frac{1}{2}}\mathbf{A}^{\frac{1}{2}}$.

### 3.3 FISHER-WEIGHTED LINEAR LAYER COMPRESSION

Building on the established connection between MVN distributions and Fisher Information, we are now ready to formulate the rank-$r$ linear layer compression theorem.

**Theorem 1.** *Let $\mathbf{W} \in \mathbb{R}^{n \times m}$ represent some parameter weights matrix of a single-layer linear neural network. Suppose that the following conditions hold.*

1. *The neural network is associated with a loss function corresponding to a Maximum Likelihood Estimation (MLE) objective (e.g., cross-entropy loss). This ensures the Hessian at convergence coincides with the Fisher Information Matrix (FIM).*

2. *The empirical FIM is approximated by a Kronecker product $I_F \approx A \otimes B$*

3. *The weights $\mathbf{W}$ are drawn from the MVN distribution $\mathcal{MN}(\mathbf{W}^\star, \mathbf{B}^{-1}, \mathbf{A}^{-1})$, where $\mathbf{W}^\star$ is the optimal weights matrix.*

*Under these conditions, the best rank-$r$ approximation that minimizes the expected increase in the loss after low-rank decomposition of $\mathbf{W}^\star$ is given by:*

$$\widehat{\mathbf{W}}_r = \mathbf{L_B}^{-\top} \widetilde{\mathbf{W}}_r \mathbf{L_A}^{-1}, \tag{7}$$

*where $\mathbf{A} = \mathbf{L_A} \mathbf{L_A}^\top$ and $\mathbf{B} = \mathbf{L_B} \mathbf{L_B}^\top$ are Cholesky factorizations, $\widetilde{\mathbf{W}} = \mathbf{L_B}^\top \mathbf{W}^\star \mathbf{L_A}$ is an auxiliary matrix, $\widetilde{\mathbf{W}}_r$ is the truncated SVD of $\widetilde{\mathbf{W}}$ of rank $r$.*

It should be noted that Condition 2 does not generally hold exactly, as neural networks often exhibit complex, non-Kronecker Hessian structure. Therefore, we treat Condition 2 as an operative Hessian approximation that enables tractable computation.

*Proof.* Under the assumption that the loss function originates from MLE, the Hessian coincides with Fisher Information at the optimal point, ensuring structured sensitivity encoding. Hence, one can replace Eq. 2 with a surrogate problem

$$\min_{\mathcal{C}} \ (\theta^\star - \mathcal{C}(\theta^\star))^\top \mathcal{I}_F (\theta^\star - \mathcal{C}(\theta^\star)) \tag{8}$$

for $\mathrm{vec}(\mathbf{W}^\star) = \theta^\star$ and $\mathrm{vec}(\mathbf{W}) = \mathcal{C}(\theta^\star)$.

Substituting $\mathcal{I}_F$ with $\mathbf{A} \otimes \mathbf{B}$ and applying Cholesky decomposition to factors $\mathbf{A}$ and $\mathbf{B}$ yields:

$$\mathrm{vec}(\mathbf{W}^\star - \mathbf{W})^\top (\mathbf{L_A} \mathbf{L_A}^\top \otimes \mathbf{L_B} \mathbf{L_B}^\top) \mathrm{vec}(\mathbf{W}^\star - \mathbf{W})$$
$$= \mathrm{vec}(\mathbf{W}^\star - \mathbf{W})^\top (\mathbf{L_A} \otimes \mathbf{L_B})(\mathbf{L_A}^\top \otimes \mathbf{L_B}^\top) \mathrm{vec}(\mathbf{W}^\star - \mathbf{W})$$
$$= \mathrm{vec}(\mathbf{L_B}^\top (\mathbf{W}^\star - \mathbf{W}) \mathbf{L_A})^\top \mathrm{vec}(\mathbf{L_B}^\top (\mathbf{W}^\star - \mathbf{W}) \mathbf{L_A})$$
$$= \left\| \mathbf{L_B}^\top (\mathbf{W}^\star - \mathbf{W}) \mathbf{L_A} \right\|_F^2 \tag{9}$$

In Section 3.2, we established that the optimal solution to this problem can be obtained via the standard SVD of the auxiliary matrix $\widetilde{\mathbf{W}}$. The final solution is found in two steps: 1) finding an optimal rank-$r$ solution to the auxiliary problem $\widetilde{\mathbf{W}}_r = \mathrm{SVD}_r(\mathbf{L_B}^\top \mathbf{W}^\star \mathbf{L_A})$, and 2) recovering the optimal solution to the original problem through the inverse transformation $\widehat{\mathbf{W}}_r = \mathbf{L_B}^{-\top} \widetilde{\mathbf{W}}_r \mathbf{L_A}^{-1}$, which yields the best rank-$r$ minimizer for Eq. 9. Consequently, the decomposition $\widehat{\mathbf{W}}_r$ presents an optimal compression $\mathcal{C}$ for Eq. 8, which in turn yields the minimal error increase in Eq. 1 for the given task defined by Eq. 2. ∎

Linear layer factorization in this case can be computed with the following expressions:

$$\mathbf{W}_1 = \hat{\mathbf{S}}_r^{\frac{1}{2}} \hat{\mathbf{V}}_r^\top \mathbf{L_A}^{-1} \in \mathbb{R}^{r \times m}, \ \mathbf{W}_2 = \mathbf{L_B}^{-\top} \hat{\mathbf{U}}_r \hat{\mathbf{S}}_r^{\frac{1}{2}} \in \mathbb{R}^{n \times r}, \tag{10}$$

where $\hat{\mathbf{S}}_r$ is the diagonal matrix of the $r$ leading singular values of the auxiliary problem.

We prove in Theorem 1 that the optimum of Problem 8 yields an SVD decomposition of the layer, weighted by the square roots of the empirical Fisher Information's factor matrices. Consequently, the procedure for obtaining this analytical optimal decomposition hinges on the efficient computation of the Fisher Information factorization for the linear layer, as elaborated in Section 4.

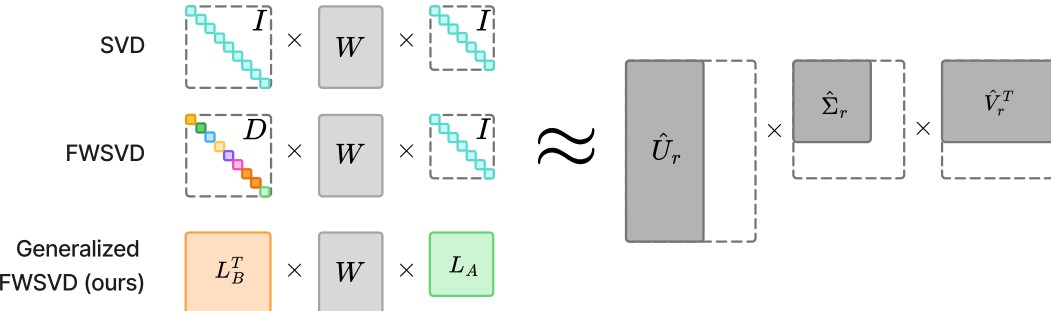

Figure 1: Generalization of the Weighted SVD frameworks. For standard SVD, the transformation matrices are identity matrices. For FWSVD, the left matrix is diagonal but not identity, and the right matrix is identity. For GFWSVD, both matrices are non-diagonal.

### 3.4 RELATIONSHIP TO PRIOR WORKS

We show that FWSVD, presented in Hsu et al. (2022), is a special case of our generalized framework. The full justification is given in Appendix A. In FWSVD, the objective minimizes a weighted reconstruction error using a diagonal matrix $\mathbf{D}$ derived from a row-wise sum of the Fisher Information. We show that this setup corresponds to a diagonal Kronecker-factored approximation of the FIM, where $\mathbf{D}$ arises naturally from minimizing the Kronecker approximation error. The resulting solution for the low-rank factors $\mathbf{W}_2, \mathbf{W}_1$ matches that of FWSVD (up to a constant), which shows that their method is a special case of our more general framework.

The connection between our generalized approach, the classical SVD and FWSVD is depicted in Figure 1. Weighted SVD approaches can be interpreted as transforming the decomposed object—here, the weight matrix—into a new space where the low-rank approximation better aligns with the target task. In this formulation, the sensitivity matrices serve as transformation matrices that reweight the importance of different directions. Under this view, vanilla SVD corresponds to using identity transformations; FWSVD applies a diagonal (but non-identity) transformation on one side while keeping the other side as identity. In contrast, our method employs full, non-diagonal transformations on both sides, capturing richer structure in the parameter space.

## 4 KRONECKER FACTORIZATION ALGORITHM VIA RANK-1 SVD

Theorem 1 and Eq. 10 state that, to obtain the provably optimal weighted SVD for a given layer, it suffices to decompose the Hessian into a Kronecker form. However, the Hessian of a linear layer scales quadratically with the layer dimension, thereby severely constraining its tractability on GPUs; this constraint is the primary motivation for existing methods to employ diagonal approximations or to use stochastic, moving-average updates for the factors. In contrast, we propose a computationally effective analytical adaptation of the Kronecker decomposition algorithm for the FIM that captures its full structure without the explicit construction of the full matrix.

Suppose that we have a linear layer of a network with a weight matrix $\mathbf{W}$ and define $\mathbf{G}_i \in \mathbb{R}^{n \times m}$ as a weight gradients $\mathcal{L}(\theta)|_{\theta=\mathbf{W}}$ on the $i$-th batch, and $g_i = \text{vec}(\mathbf{G}_i) \in \mathbb{R}^{n \cdot m}$ - its flattening version. Then, Fisher Information $\mathcal{I}_F(\theta)$ can be defined as an empirical mean over all batches in a dataset $D$:

$$\mathcal{I}_F(\theta^\star) = \mathbb{E}\left[gg^\top\right] = \frac{1}{|D|} \sum_{i=1}^{|D|} g_i g_i^\top. \tag{11}$$

Kronecker product approximation is obtained by solving minimization problem:

$$\min \|\mathcal{I}_F - \mathbf{A} \otimes \mathbf{B}\|_{\text{F}} \tag{12}$$

This minimization problem is equivalent to finding the best rank-1 approximation of a permuted Fisher matrix $\tilde{\mathcal{I}}_F = \mathcal{R}\mathcal{I}_F \in \mathbb{R}^{m^2 \times n^2}$, as established by Van Loan & Pitsianis (1993). Specifically,

the singular vectors associated with the largest singular value of $\tilde{\mathcal{I}}_F$ yield the optimal factors $\mathbf{A}$ and $\mathbf{B}$. We summarize this efficient decomposition procedure in Algorithm 1.

---

**Algorithm 1** Compute Kronecker Factors via Rank-1 SVD

---

**Require:** List of gradients $\{g_i\}_{i=1}^{|D|}$, $|D|$ – number of batches
1: $\mathcal{I}_F \leftarrow \frac{1}{|D|} \sum_{i=1}^{|D|} g_i g_i^T$
2: $\tilde{\mathcal{I}}_F \leftarrow \mathcal{R}\mathcal{I}_F \leftarrow \frac{1}{|D|} \sum_{i=1}^{|D|} \mathbf{G_i} \otimes \mathbf{G_i}$
3: $(u, \sigma, v^\top) \leftarrow$ Leading singular triplet $\qquad\qquad\qquad\qquad \triangleright$ Truncated SVD
4: $b \leftarrow u \cdot \sigma$ $\qquad\qquad\qquad\qquad\qquad\qquad\qquad\qquad\qquad\qquad \triangleright b = \text{vec}(\mathbf{B})$
5: $a \leftarrow v$ $\qquad\qquad\qquad\qquad\qquad\qquad\qquad\qquad\qquad\qquad\qquad \triangleright a = \text{vec}(\mathbf{A})$
6: $\mathbf{B} \leftarrow \text{reshape}(b, (m, m))$
7: $\mathbf{A} \leftarrow \text{reshape}(a, (n, n))$
8: **return** $(\mathbf{B}, \mathbf{A})$

---

## 4.1 EFFICIENT RANK-1 COMPUTATION

The primary computational bottleneck of Algorithm 1 arises in performing SVD on the matrix $\tilde{\mathcal{I}}_F$.

Standard SVD is computationally intractable for large matrices, so we employ truncated SVD using the Lanczos method (Lanczos, 1950), which avoids explicit matrix construction and requires only the ability to multiply the matrix with a vector from the left or right. Even in this setting, aggregating the full second-moment gradient information across all batch samples is computationally expensive.

We can show (see Appendix B) that permuted $\mathcal{I}_F$ for $i$-th batch can be defined as the Kronecker product of the corresponding gradient matrices:

$$\tilde{\mathcal{I}}_F = \frac{1}{|D|} \sum_{i=1}^{|D|} \mathbf{G}_i \otimes \mathbf{G}_i. \tag{13}$$

If we multiply this matrix $\tilde{\mathcal{I}}_F$ by a vector $z$ from left, it will yield:

$$\tilde{\mathcal{I}}_F z = \frac{1}{k} \left( \sum_{i=1}^{k} \mathbf{G}_i \otimes \mathbf{G}_i \right) z = \frac{1}{|D|} \left( \sum_{i=1}^{|D|} \mathbf{G}_i \otimes \mathbf{G}_i \right) \mathbf{Z} = z, \text{ where } z = \text{vec}(\mathbf{Z}), \mathbf{Z} \in \mathbb{R}^{n \times n}. \tag{14}$$

Using property of the Kronecker product $(\mathbf{K} \otimes \mathbf{L}) \text{vec}(\mathbf{C}) = \text{vec}(\mathbf{K}^\top \mathbf{C} \mathbf{L})$ we reduce the matrix-vector multiplication to a sequence of matrix multiplications:

$$\tilde{\mathcal{I}}_F z = \frac{1}{|D|} \sum_{i=1}^{|D|} \text{vec}(\mathbf{G}_i^\top \mathbf{Z} \mathbf{G}_i) \tag{15}$$

The derivation for right-side multiplication is analogous (see Appendix C).

These operations allow us to efficiently approximate the Fisher matrix for LLM layers at practical batch sizes. As stated in Step 1 of Algorithm 1 and Eq. 13, neither the full Fisher matrix $\mathcal{I}_F$ nor its permuted form $\tilde{\mathcal{I}}_F$ is ever constructed. Instead, we reduce all computations to operations scaled to the layer size.

## 4.2 THEORETICAL TIME COMPLEXITY OF THE PROPOSED RANK-1 COMPUTATION

The time complexity of computing the truncated SVD of the matrix $\tilde{\mathcal{I}}_F \in \mathbb{R}^{m^2 \times n^2}$ consists of the matrix-vector multiplications and the orthogonalization and has a cost of $\mathcal{O}\left(m^2 n^2\right)$. However, using the structured formulation from Eq. 15, where left matrix-vector products are implemented via multiplications with matrices $\mathbf{G}_i^\top \in \mathbb{R}^{m \times n}$, $\mathbf{Z} \in \mathbb{R}^{n \times n}$, and $\mathbf{G}_i \in \mathbb{R}^{n \times m}$, the overall complexity is reduced to $\mathcal{O}\left(mn^2 + m^2 n\right)$. Applying analogous reasoning to the right matrix-vector products (see Eq. 30) one can yield the same complexity.

### 4.3 EMPIRICAL TIME COMPLEXITY AND APPLICATION TO LLMS

In the context of LLM, accelerated Hessian decomposition Algorithm 1 is practical as long as a Hessian of a single layer can be decomposed quickly. This is a relevant bottleneck that comes from the fact that the Hessian size grows quadratically with the layer size and contains more than $10^{12}$ elements. In Table 1, we report the empirical Hessian decomposition times for single linear layer in different LLMs and prove that this accelerated algorithm is tractable on many transformer models.

For the entire LLM, layers are processed independently and factor computation can be parallelized, so the runtime scales as:

$$\text{Total time} = \frac{\text{time per layer} \times \text{number of layers}}{\text{number of workers}}.$$

For example, Llama 2 7B model that has 224 linear layers, can be compressed in approximately 3.5 hours on 3 A100 GPUs. The VRAM constraints are minimal: the peak footprint occurs only during gradient accumulation, which we manage by processing layers sequentially by freezing and unfreezing modules iteratively. That way we fit within standard VRAM limits.

Table 1: Runtime for computing Kronecker factors of single linear layer on GPU.

| Model | Params in layer | Params in Hessian | Decomp. time (s) |
|---|---|---|---|
| BERT | $2.3\times10^6$ | $5.5\times10^{12}$ | 43 |
| Llama 2 7B | $45\times10^6$ | $2.0\times10^{15}$ | 183 |
| Llama 3.1 8B | $58\times10^6$ | $3.4\times10^{15}$ | 249 |
| Llama 2 13B | $70.8\times10^6$ | $4.9\times10^{15}$ | 313 |

## 5 NUMERICAL EXPERIMENTS

To validate our theoretical contributions, we conduct extensive numerical experiments on several transformer architectures: the encoder-only BERT model (Devlin et al., 2019) and the recent open-weights decoder-only LLMs Llama 2 (Touvron et al., 2023) and Llama 3.1 (Team, 2024). Our goal is to demonstrate the practical benefits of GFWSVD in low-rank compression under fine-tuning and evaluation protocols.

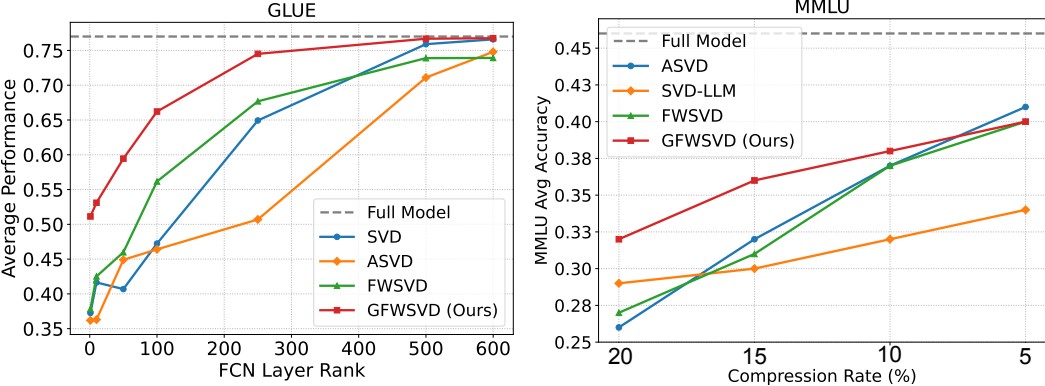

Figure 2: Macro-averaged GLUE performance of BERT model for different compression ranks.

Figure 3: Average MMLU performance of Llama 2 model for different compression rates.

### 5.1 COMPRESSING THE TRANSFORMER ENCODER

In our experiments, we follow the "fine-tune then compress" pipeline, similar to FWSVD (Hsu et al., 2022). We begin by fine-tuning a pre-trained checkpoint[2] of the BERT-base model on a specific downstream GLUE task. Optimal fine-tuning hyperparameters (e.g., learning rate, batch size) are selected for each task using the Optuna framework (Akiba et al., 2019). During this stage, we also collect gradients to construct the FIM $\mathcal{I}_F$ and compute its Kronecker decomposition as described in Section 4.

Using the resulting Cholesky factors $\mathbf{L_A}$ and $\mathbf{L_B}$, we uniformly compress the fully connected layers of BERT by factorizing them into two smaller layers, following the method detailed in Section 3.1.

---

[2] https://huggingface.co/google-bert/bert-base-uncased

The chosen layer-wise ranks and the resulting overall compression rate of the model are summarized in Table 2. We reproduce the ASVD method using the original authors' code. For FWSVD, we incorporate the newly constructed FIM into the compression process.

We show average compression results in Table 2 and Figure 2, extended results are in Appendix F in Table 11. On most of the GLUE tasks and considered compression ranks, our proposed GFWSVD approach consistently outperforms both FWSVD and SVD, with particularly strong gains at lower ranks. While ASVD exhibits relatively poor performance on several tasks (QQP, QNLI), it occasionally surpasses GFWSVD — notably on SST2 under aggressive compression.

Table 2: Macro-averaged GLUE performance of BERT for different compression ranks. Best results for each rank are in **bold**. We denote compression ratio as $1 - \frac{\text{compressed Model}}{\text{original Model}}$.

| Method / Rank | 600 | 500 | 250 | 100 | 50 | 10 | 1 |
| Compression ratio | 1% | 8% | 23% | 33% | 34% | 39% | 40% |
| --- | --- | --- | --- | --- | --- | --- | --- |
| SVD | **0.77** | 0.76 | 0.65 | 0.47 | 0.41 | 0.42 | 0.37 |
| ASVD | 0.75 | 0.71 | 0.51 | 0.46 | 0.45 | 0.36 | 0.36 |
| FWSVD | 0.74 | 0.74 | 0.68 | 0.56 | 0.46 | 0.43 | 0.38 |
| GFWSVD (Ours) | **0.77** | **0.77** | **0.75** | **0.66** | **0.59** | **0.53** | **0.51** |

## 5.2 COMPRESSING THE TRANSFORMER DECODER

We evaluate our approach on the decoder-only models Llama 2 7B[3] and Llama 3.1 8B[4]. Since GFWSVD is purely analytical – containing no stochastic steps – we benchmark it against several competitive baselines of the same class of methods: diagonal FI-based low-rank approximation method FWSVD (Hsu et al., 2022), two activation-based methods – ASVD (Yuan et al., 2023) and SVD-LLM (Wang et al., 2025c), and per-layer relation-aware Basis Sharing (Wang et al., 2024). Notably, ASVD and SVD-LLM both rely on activation-based weighting to gauge parameter importance, while Basis Sharing relies on correlations across layers in the entire model. In contrast, FWSVD and our GFWSVD rely solely on gradient information, treating each layer as independent.

We measure perplexity on WikiText 2 (Merity et al., 2017) and PTB (Marcus et al., 1993) datasets, 5-shot reasoning performance on the MMLU benchmark (Hendrycks et al., 2021) and 0-shot performance on OpenBookQA (Banerjee & Baral, 2020), WinoGrande (Sakaguchi et al., 2021), HellaSwag (Zellers et al., 2019), PIQA (Bisk et al., 2020), ARC-E and ARC-C (Clark et al., 2018). Following prior works on low-rank approximation of LLMs (Wang et al., 2025c; Yuan et al., 2023), we test several compression setups, removing from 5% to 50% of original parameters.

Table 3: Performance of the Llama 3.1 8B Instruct compressed by various methods under compression ratios from 20% to 50% on WikiText-2, PTB, and six common sense reasoning datasets. Lower is better for perplexity ($\downarrow$), higher is better for accuracy ($\uparrow$). We denote compression ratio as $1 - \frac{\text{compressed Model}}{\text{original Model}}$.

| METHOD | WikiText$\downarrow$ | PTB$\downarrow$ | C. Ratio | ARC-C$\uparrow$ | ARC-E$\uparrow$ | HellaSwag$\uparrow$ | PIQA$\uparrow$ | WinoG.$\uparrow$ | OpenBook$\uparrow$ | AVG$\uparrow$ |
| --- | --- | --- | --- | --- | --- | --- | --- | --- | --- | --- |
| Full model | **7.20** | **11.50** | 100% | **0.52** ± 0.01 | **0.81** ± 0.01 | **0.59** ± 0.01 | **0.79** ± 0.01 | **0.73** ± 0.01 | **0.35** ± 0.02 | **0.63** |
| FWSVD | 354 | 864 | | 0.21 ± 0.01 | 0.38 ± 0.01 | 0.20 ± 0.01 | 0.60 ± 0.01 | 0.52 ± 0.01 | 0.17 ± 0.02 | 0.35 |
| ASVD | 145 | 1672 | 20% | 0.21 ± 0.01 | 0.33 ± 0.01 | 0.27 ± 0.01 | 0.61 ± 0.01 | 0.54 ± 0.01 | 0.15 ± 0.02 | 0.35 |
| Basis Sharing | **18.54** | 90.05 | | 0.34 ± 0.01 | **0.68** ± 0.01 | 0.42 ± 0.01 | 0.70 ± 0.01 | **0.65** ± 0.01 | **0.35** ± 0.02 | 0.52 |
| GFWSVD (Ours) | 22.57 | **42.40** | | **0.35** ± 0.01 | **0.68** ± 0.01 | **0.45** ± 0.01 | **0.75** ± 0.01 | 0.63 ± 0.01 | 0.33 ± 0.02 | **0.53** |
| FWSVD | 4372 | 6824 | | 0.21 ± 0.01 | 0.30 ± 0.01 | 0.26 ± 0.01 | 0.57 ± 0.01 | 0.51 ± 0.01 | 0.14 ± 0.02 | 0.33 |
| ASVD | 1456 | 4232 | 30% | 0.22 ± 0.01 | 0.30 ± 0.01 | 0.25 ± 0.01 | 0.58 ± 0.01 | 0.52 ± 0.01 | 0.16 ± 0.02 | 0.34 |
| Basis Sharing | **32** | 286 | | 0.29 ± 0.01 | 0.52 ± 0.01 | **0.43** ± 0.01 | 0.63 ± 0.01 | **0.60** ± 0.01 | **0.31** ± 0.02 | 0.46 |
| GFWSVD (Ours) | 35 | 58 | | **0.33** ± 0.01 | **0.61** ± 0.01 | 0.42 ± 0.01 | **0.71** ± 0.01 | 0.58 ± 0.01 | 0.23 ± 0.02 | **0.48** |
| FWSVD | 11072 | 15376 | | 0.21 ± 0.01 | 0.27 ± 0.01 | 0.26 ± 0.01 | 0.54 ± 0.01 | 0.48 ± 0.01 | 0.16 ± 0.02 | 0.32 |
| ASVD | 2992 | 13193 | 40% | 0.23 ± 0.01 | 0.27 ± 0.01 | 0.26 ± 0.01 | 0.55 ± 0.01 | 0.49 ± 0.01 | 0.15 ± 0.02 | 0.33 |
| Basis Sharing | 78 | 1083 | | 0.24 ± 0.01 | 0.39 ± 0.01 | **0.33** ± 0.01 | 0.56 ± 0.01 | **0.56** ± 0.01 | **0.28** ± 0.02 | **0.39** |
| GFWSVD (Ours) | **69** | **101** | | **0.25** ± 0.01 | **0.41** ± 0.01 | 0.32 ± 0.01 | **0.61** ± 0.01 | 0.55 ± 0.01 | 0.22 ± 0.02 | **0.39** |
| FWSVD | 18992 | 23088 | | 0.20 ± 0.01 | 0.27 ± 0.01 | 0.26 ± 0.01 | 0.50 ± 0.01 | 0.51 ± 0.01 | 0.15 ± 0.02 | 0.31 |
| ASVD | 4039 | 46189 | 50% | 0.22 ± 0.01 | 0.26 ± 0.01 | 0.26 ± 0.01 | 0.50 ± 0.01 | 0.48 ± 0.01 | 0.13 ± 0.02 | 0.31 |
| Basis Sharing | 203 | 3506 | | 0.23 ± 0.01 | 0.30 ± 0.01 | **0.29** ± 0.01 | 0.52 ± 0.01 | 0.53 ± 0.01 | **0.26** ± 0.02 | 0.35 |
| GFWSVD (Ours) | **176** | **501** | | **0.24** ± 0.01 | **0.31** ± 0.01 | 0.28 ± 0.01 | **0.55** ± 0.01 | **0.54** ± 0.01 | 0.22 ± 0.02 | **0.36** |

---

[3] https://huggingface.co/meta-llama/Llama-2-7b-chat-hf
[4] https://huggingface.co/meta-Llama/Llama-3.1-8B-Instruct

Following standard practice in post-training LLM compression methods (Wang et al., 2025c; Yuan et al., 2023), we use a randomly sampled set of sentences as calibration data to generate gradients for further obtaining the factor matrices. For calibration data, we choose the FineWeb dataset (Penedo et al., 2024) due to its high quality and diversity, and collect gradients on a random subsample of size 1024. These gradients are then used to obtain $\mathbf{L_A}$ and $\mathbf{L_B}$, as well as the data needed for FWSVD. As in LLMs, uniform layer compression can disproportionately degrade performance by over-compressing critical layers and under-utilizing redundancy in less sensitive ones, so it is essential for each method to use a compression configuration that accounts for layer sensitivity. For both ASVD and SVD-LLM, we used the corresponding code released by the authors and re-ran the necessary compression pipelines for our checkpoint with all hyperparameters set to default values. For our approach, we adopted the method of per-layer importance scores as described in the ASVD work.

Table 4: Performance of the LLaMA 2 7B Chat compressed by various methods under compression ratios from 20% to 50% on WikiText-2 and six common sense reasoning datasets. Lower is better for perplexity ($\downarrow$), higher is better for accuracy ($\uparrow$).

| METHOD | WikiText↓ | C. Ratio | ARC-C↑ | ARC-E↑ | HellaSwag↑ | PIQA↑ | WinoG.↑ | OpenBook↑ | AVG↑ |
|---|---|---|---|---|---|---|---|---|---|
| Full model | **6.94** | 100% | **0.44** ± 0.01 | **0.73** ± 0.01 | **0.58** ± 0.01 | **0.76** ± 0.01 | **0.67** ± 0.01 | **0.33** ± 0.02 | **0.59** |
| FWSVD | 66.18 | | 0.24 ± 0.01 | 0.48 ± 0.01 | 0.38 ± 0.01 | 0.64 ± 0.01 | 0.58 ± 0.01 | 0.18 ± 0.02 | 0.42 |
| ASVD | 18.33 | | 0.27 ± 0.01 | 0.51 ± 0.01 | 0.39 ± 0.01 | 0.68 ± 0.01 | 0.61 ± 0.01 | 0.22 ± 0.02 | 0.45 |
| SVD-LLM | 12.10 | 20% | 0.29 ± 0.01 | **0.66** ± 0.01 | 0.40 ± 0.01 | 0.66 ± 0.01 | **0.61** ± 0.01 | 0.23 ± 0.02 | 0.48 |
| Basis Sharing | **11.1** | | 0.31 ± 0.01 | 0.65 ± 0.01 | 0.42 ± 0.01 | 0.68 ± 0.01 | **0.61** ± 0.01 | **0.27** ± 0.02 | 0.49 |
| GFWSVD (Ours) | **11.1** | | **0.33** ± 0.01 | 0.62 ± 0.01 | **0.47** ± 0.01 | **0.74** ± 0.01 | **0.61** ± 0.01 | 0.25 ± 0.02 | **0.50** |
| FWSVD | 2572 | | 0.24 ± 0.01 | 0.32 ± 0.01 | 0.27 ± 0.01 | 0.58 ± 0.01 | 0.51 ± 0.01 | 0.17 ± 0.02 | 0.35 |
| ASVD | 97.68 | | 0.21 ± 0.01 | 0.31 ± 0.01 | 0.29 ± 0.01 | 0.63 ± 0.01 | 0.54 ± 0.01 | 0.15 ± 0.02 | 0.36 |
| SVD-LLM | 18.29 | 30% | 0.25 ± 0.01 | 0.52 ± 0.01 | 0.34 ± 0.01 | 0.62 ± 0.01 | 0.55 ± 0.01 | 0.22 ± 0.02 | 0.42 |
| Basis Sharing | 15.40 | | 0.27 ± 0.01 | **0.58** ± 0.01 | 0.38 ± 0.01 | **0.63** ± 0.01 | **0.58** ± 0.01 | **0.26** ± 0.02 | **0.45** |
| GFWSVD (Ours) | **13.92** | | **0.28** ± 0.01 | 0.56 ± 0.01 | 0.40 ± 0.01 | 0.63 ± 0.01 | 0.58 ± 0.01 | 0.20 ± 0.02 | 0.44 |
| FWSVD | 9286 | | 0.23 ± 0.01 | 0.26 ± 0.01 | 0.25 ± 0.01 | 0.48 ± 0.01 | 0.45 ± 0.01 | 0.16 ± 0.02 | 0.31 |
| ASVD | 2992 | | 0.22 ± 0.01 | 0.26 ± 0.01 | 0.26 ± 0.01 | 0.49 ± 0.01 | 0.49 ± 0.01 | 0.16 ± 0.02 | 0.31 |
| SVD-LLM | 25.16 | 40% | 0.26 ± 0.01 | 0.45 ± 0.01 | 0.30 ± 0.01 | 0.55 ± 0.01 | 0.54 ± 0.01 | 0.19 ± 0.02 | 0.38 |
| Basis Sharing | 17.26 | | 0.21 ± 0.01 | 0.46 ± 0.01 | 0.32 ± 0.01 | 0.58 ± 0.01 | 0.55 ± 0.01 | **0.19** ± 0.02 | 0.39 |
| GFWSVD (Ours) | **16.70** | | **0.27** ± 0.01 | **0.48** ± 0.01 | **0.33** ± 0.01 | **0.64** ± 0.01 | **0.57** ± 0.01 | 0.17 ± 0.02 | **0.41** |
| FWSVD | 36578 | | 0.22 ± 0.01 | 0.25 ± 0.01 | 0.25 ± 0.01 | 0.52 ± 0.01 | 0.50 ± 0.01 | 0.17 ± 0.02 | 0.32 |
| ASVD | 16896 | | 0.21 ± 0.01 | 0.25 ± 0.01 | 0.26 ± 0.01 | 0.53 ± 0.01 | 0.49 ± 0.01 | 0.16 ± 0.02 | 0.32 |
| SVD-LLM | 56.72 | 50% | 0.21 ± 0.01 | 0.33 ± 0.01 | 0.26 ± 0.01 | 0.54 ± 0.01 | 0.50 ± 0.01 | 0.12 ± 0.02 | 0.33 |
| Basis Sharing | **35.12** | | 0.20 ± 0.01 | **0.36** ± 0.01 | **0.30** ± 0.01 | **0.55** ± 0.01 | 0.50 ± 0.01 | **0.15** ± 0.02 | **0.34** |
| GFWSVD (Ours) | 37.80 | | **0.22** ± 0.01 | 0.28 ± 0.01 | 0.26 ± 0.01 | **0.55** ± 0.01 | **0.51** ± 0.01 | **0.15** ± 0.02 | 0.33 |

Tables 5.2 and 4 show that on LLaMA-2 7B and LLaMA-3 8B, and for compression levels up to 50%, GFWSVD substantially outperforms methods that decompose layers independently, both diagonal FWSVD and activation-aware ASVD. If we compare GFWSVD, which relies on parameter correlations within a layer, with Basis Sharing, which captures correlations across layers, GFWSVD outperforms Basis Sharing on LLaMA-3 8B at all compression levels and surpasses it on LLaMA-2 7B at 20% and 40% compression. This difference likely stems from stronger inter-layer correlations in the instruction-tuned LLaMA-3 8B model. We also observe that GFWSVD and Basis Sharing behave differently across tasks while maintaining consistent trends across compression ratios. For example, on PIQA, GFWSVD often surpasses Basis Sharing by nearly 10%, whereas on OpenBookQA the opposite pattern emerges. This suggests that different types of structural dependencies within the model—parameter-level versus inter-layer relationships—benefit different categories of tasks.

More fine-grained compression results at 5–20% and MMLU evaluation are provided in Appendix D and Figure 3. There, we show that as the compression ratio decreases, the relative importance of diagonal Fisher information grows, and GFWSVD increasingly outperforms both FWSVD and ASVD.

Throughput and FLOP results for compressed models are provided in Appendix E.

## 5.3 METHOD POSITIONING AND APPLICABILITY

GFWSVD is essentially a standard SVD reweighted on full second-order model loss information which made them (by Theorem 1) provably optimal for the given LLM. We compute the factors for optimal reweighting analytically via a single decomposition of the Hessian, which makes FWSVD training-free and keeps its computational path close to standard SVD. As noted in the Section 2, several SVD-based pipelines optimize decomposition parameters during training and thereby achieve stronger compression. By design, GFWSVD can be seamlessly integrated into any such pipeline as a drop-in replacement for standard SVD, since its factors are computed once before training. For example, post-training compression pipeline Dobi-SVD fine-tune singular values of decomposition. We replaced standard SVD with GFWSVD in the Dobi-SVD pipeline and trained the LLaMA 2 7B model for 20 epochs using the hyperparameters provided in the official Dobi-SVD implementation[5] with remapping enabled. We evaluated resulting models on ARC-C, ARC-E, PIQA and HellaSwag. The results are reported in Table 5. Dobi-GFWSVD method maintains competitive downstream performance at 20% compression and shows only a moderate degradation ($\sim 10\%$ drop in HellaSwag accuracy) at a stronger 40% compression ratio, outperforming the original Dobi-SVD baseline in all cases. Dobi-GFWSVD exhibits only a 3% perplexity increase at 20% compression and remains competitive even at 40%.

For comparison with non–structural-approximation approaches, we also include YAQA (Tseng et al., 2025) in baselines. YAQA is a quantization method that, like our GFWSVD, leverages second-order loss information. As expected, quantization delivers stronger accuracy and less perpelxity at these compression ratios.

It is also important to emphasize the trade-off between performance and

Table 5: Performance of the LLaMA-2 7B Chat model compressed with the Dobi-SVD and Dobi-GFWSVD pipelines at 20% and 40% compression. YAQA is a second-order quantization method.

| METHOD | C. Ratio | Wiki-2↓ | PTB↓ | ARC-E↑ | ARC-C↑ | PIQA↑ | HSwag↑ |
|---|---|---|---|---|---|---|---|
| Full model | 0% | 6.94 | 25.75 | 0.73 | 0.44 | 0.78 | 0.57 |
| Dobi-SVD | | 7.75 | 26.11 | 0.71 | 0.40 | 0.76 | 0.55 |
| Dobi-GFWSVD (Ours) | 20% | **7.56** | **25.95** | **0.72** | **0.42** | **0.77** | **0.55** |
| YAQA *(quant.)* | | *6.99* | *–* | *0.73* | *0.44* | *0.78* | *0.56* |
| Dobi-SVD | | 10.56 | 41.70 | 0.58 | 0.32 | 0.69 | 0.32 |
| Dobi-GFWSVD (Ours) | 40% | **10.29** | **38.56** | **0.60** | **0.33** | **0.69** | **0.32** |
| YAQA *(quant.)* | | *8.14* | *–* | *0.66* | *0.41* | *0.77* | *0.53* |

training efficiency. GFWSVD is fully analytical and requires only 3.5 hours on 3 GPUs (including 45 calibration steps and factorization), whereas Dobi-SVD and Dobi-GFWSVD fine-tuning take roughly 20 hours on 8 GPUs.

## 6 CONCLUSION AND FUTURE WORK

We introduced **Generalized Fisher-Weighted SVD (GFWSVD)**, a low-rank second-order compression method that leverages the full Fisher Information Matrix through a scalable Kronecker decomposition. Unlike previous approaches, GFWSVD captures parameter correlations and yields a factorization *provably optimal* within its class (Theorem 1). Our results on both encoder-only (BERT on GLUE) and decoder-only (LLaMA family on reasoning datasets) show that GFWSVD consistently outperforms diagonal Fisher- and activation-based SVD approaches, particularly at higher compression rates. As for the Basis Sharing method, which employs cross-layer correlation information, our approach outperforms it on LLaMA-3 8B and partially outperforms it on LLaMA-2 7B.

Crucially, the method is entirely analytical and does *not* require stochastic optimization or iterative retraining, making it lightweight and reproducible. The tractable algorithm for computing full Kronecker factors makes this work an important step toward practical, curvature-aware post-training compression of large language models.

GFWSVD highlights the critical role of accurate FIM computation in compression. While our approach performs well empirically, its reliance on a rank-1 Kronecker approximation of the Fisher matrix may oversimplify important structure. Future work could explore higher-rank Kronecker series to capture richer information, and extend the method to model cross-layer dependencies, potentially improving performance by leveraging transitive correlations across the network.

---

[5]https://github.com/wangqinsi1/Dobi-SVD

## ETHICS STATEMENT

This work focuses on methods for improving the efficiency and practicality of post-training low-rank compression of large language models using second-order information. Our research does not involve human subjects, personally identifiable information, or other sensitive data. All experiments are carried out on publicly available models (BERT, Llama) and widely used benchmarks (GLUE, MMLU), ensuring transparency and reproducibility. We do not release any new datasets containing private or proprietary information. The proposed methods are intended to reduce the computational cost and energy consumption of deploying large models, which we view as a positive contribution to sustainability. We are not aware of any direct negative societal impacts; however, as with any model compression technique, improved efficiency may lower the barrier to deploying large models in contexts where misuse is possible. We therefore encourage responsible use of these methods in accordance with the ICLR Code of Ethics.

## REPRODUCIBILITY STATEMENT

We have made every effort to ensure reproducibility of our results. The full description of the proposed method, including theoretical assumptions and proofs, is provided in the main text and Appendix. We conduct all experiments on a 4 NVIDIA A100 GPU with latest CUDA drivers using Python 3.12. The reference implementation of the Algorithm 1 as well as all experimental pipelines are available in an anonymous repository[6].

---

[6]https://anonymous.4open.science/r/FisherKronecker-B4F0

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

## A APPENDIX: SPECIAL CASE OF DIAGONAL FISHER INFORMATION MATRIX

In this section, we show that FWSVD, presented in (Hsu et al., 2022), is a special case of our generalized approach.

In the work of (Hsu et al., 2022), authors propose to minimize the following objective:

$$\min_{\mathbf{W_1}, \mathbf{W_2}} \|\mathbf{DW}^\star - \mathbf{DW_2W_1}\|_F^2 \qquad (16)$$

where $\mathbf{D}$ is the diagonal matrix $\sqrt{\mathrm{diag}\left(\mathbb{E}[\mathbf{GG}^\top]\right)}$. Specifically, $\mathbf{D}_{i,i} = \sqrt{\sum_{j=1}^m \mathbb{E}(\mathbf{G}_{i,j})^2}$.

Similarly to 12, we approximate the Fisher Information with a Kronecker product of identity matrix $\mathbf{I}_m$ and some diagonal matrix $\tilde{\mathbf{D}}$. As described further in Section 4 and Appendix A, under the permutation $\mathcal{R}$, the problem

$$\min_{\mathbf{D}} \left\|\mathbf{I}_F - \mathbf{I}_m \otimes \tilde{\mathbf{D}}\right\|_{\mathrm{F}} \qquad (17)$$

reduces to minimization of the expression

$$\min_{\mathbf{d}} \left\|\mathbb{E}[\mathbf{G} \otimes \mathbf{G}] - (\mathbf{I}_n \odot \mathbf{I}_n)d \cdot \mathrm{vec}(\mathbf{I}_m)^\top\right\|_{\mathrm{F}} \qquad (18)$$

where $\odot$ is a Khatri-Rao product (column-wise Kronecher product) and $\cdot$ is a vector outer product; $d$ is a vector diagonal of $\tilde{\mathbf{D}}$; $\mathbb{E}[\mathbf{G} \otimes \mathbf{G}]$ is a permuted Fisher Information matrix $\tilde{\mathbf{I}}_{\mathbf{F}}$, defined in Eq 13.

For simplicity, we will use a shorter notation. Let $\mathbf{E} = \mathbb{E}[\mathbf{G} \otimes \mathbf{G}]$, $\mathbf{Z} = \mathbf{I}_n \odot \mathbf{I}_n$, $v = \mathrm{vec}(\mathbf{I}_m)$. Then, the problem 18 is equivalent to

$$\min_{\mathbf{d}} \left\| \mathbf{Z}d \cdot v^\top - \mathbf{E} \right\|_F \tag{19}$$

Applying first-order optimality conditions yields:

$$\langle \mathbf{Z}\delta d \cdot v^\top, \mathbf{Z}d \cdot v^\top - \mathbf{E} \rangle = 0$$
$$\langle \delta d \cdot v^\top, \mathbf{Z}^\top \mathbf{Z}d \cdot v^\top - \mathbf{Z}^\top \mathbf{E} \rangle = 0$$
$$\langle \delta d, \mathbf{Z}^\top \mathbf{Z}d \cdot v^\top v - \mathbf{Z}^\top \mathbf{E}v \rangle = 0$$

Since $\mathbf{Z}^\top \mathbf{Z} = \mathbf{I}_n$, $v^\top v = \|v\|_2^2 = \|\mathrm{vec}(\mathbf{I}_m)\|_2^2 = m$ , we have:

$$d = \frac{1}{m}(\mathbf{I}_n \odot \mathbf{I}_n)^\top \mathbb{E}[\mathbf{G} \otimes \mathbf{G}]\mathrm{vec}(\mathbf{I}_m) = \frac{1}{m}(\mathbf{I}_n \odot \mathbf{I}_n)^\top \mathrm{vec}(\mathbb{E}[GG^\top]) = \frac{1}{m}\mathrm{diag}(\mathbb{E}[GG^\top]) \tag{20}$$

Thus, diagonal matrix $\tilde{\mathbf{D}}$ from Kronecker product approximation problem 17 equals square of matrix $\mathbf{D}$ from the FWSVD formulation 16 up to the constant $\frac{1}{m}$.

We apply Theorem 1 to find factors $\mathbf{W}_2$, $\mathbf{W}_1$ for the obtained approximation $\mathbf{I}_F = \mathbf{I}_m \otimes \tilde{\mathbf{D}}$:

$$\mathbf{W}_2 = \sqrt{\tilde{\mathbf{D}}}^{-1} \hat{\mathbf{U}}_r \sqrt{\hat{\mathbf{S}}_r} = \mathbf{D}^{-1}\hat{\mathbf{U}}_r \sqrt{\hat{\mathbf{S}}_r}, \mathbf{W}_1 = \sqrt{\hat{\mathbf{S}}_r}\hat{\mathbf{V}}_r^\top \tag{21}$$

where $\hat{\mathbf{U}}_r \hat{\mathbf{S}}_r \hat{\mathbf{V}}_r^\top$ is r-rank SVD of $\sqrt{\tilde{\mathbf{D}}}\mathbf{W}^\star = \mathbf{D}\mathbf{W}^\star$. This is the same solution that minimizes the problem 16 from FWSVD paper (Hsu et al., 2022). Consequently, FWSVD approach is a special case of diagonal Kronecker product approximation of Fisher Information.

## B  APPENDIX: ADDITIONAL EXPLANATIONS FOR KRONECKER DECOMPOSITION ADAPTATION

Let's show that the permuted $\mathcal{I}_F$ in the Kronecker decomposition algorithm can be expressed as the Kronecker product of the corresponding gradient matrices.

We start with the empirical Fisher information matrix defined as $\mathcal{I}_F = \frac{1}{|D|}\sum_{i=1}^{|D|} g_i g_i^\top$ and its reordered version:

$$\tilde{\mathcal{I}}_F = \mathcal{R}\mathcal{I}_F \tag{22}$$

Using the identity

$$\mathrm{vec}(\boldsymbol{g}_i \boldsymbol{g}_i^\top) = \boldsymbol{g}_i \otimes \boldsymbol{g}_i,$$

we obtain:

$$\mathrm{vec}(\mathcal{I}_F) = \frac{1}{|D|}\sum_{i=1}^{|D|} \mathrm{vec}(\boldsymbol{g}_i \boldsymbol{g}_i^\top) = \frac{1}{|D|}\sum_{i=1}^{|D|}(\boldsymbol{g}_i \otimes \boldsymbol{g}_i). \tag{23}$$

Let $\mathcal{P} \in \mathbb{R}^{(ab)^2 \times (ab)^2}$ be the unique permutation matrix such that for any matrices $\mathbf{A}, \mathbf{B} \in \mathbb{R}^{a \times b}$:

$$\mathcal{P} \cdot \mathrm{vec}(\mathbf{A} \otimes \mathbf{B}) = (\mathrm{vec}(\mathbf{A}) \otimes \mathrm{vec}(\mathbf{B})). \tag{24}$$

In our case $\mathcal{P}$ can be defined through the commutation matrix $\mathbf{K}_{mn}$ and identity matrices $\mathbf{I}_n$ and $\mathbf{I}_m$:

$$\mathbf{P} := \mathbf{I}_n \otimes \mathbf{K}_{mn} \otimes \mathbf{I}_m, \qquad \mathbf{K}_{mn}^\top = \mathbf{K}_{nm} \tag{25}$$

Using this , we can write:

$$\mathcal{P} \cdot \text{vec}(\mathbf{G}_i \otimes \mathbf{G}_i) = \text{vec}(\mathbf{G}_i) \otimes \text{vec}(\mathbf{G}_i). \tag{26}$$

Therefore, the vectorized Fisher information becomes:

$$\text{vec}(\mathcal{I}_F) = \frac{1}{|D|} \sum_{i=1}^{|D|} \mathcal{P} \cdot \text{vec}(\mathbf{G}_i \otimes \mathbf{G}_i) = \mathcal{P} \cdot \text{vec}\left(\frac{1}{|D|} \sum_{i=1}^{|D|} (\mathbf{G}_i \otimes \mathbf{G}_i)\right) = \mathcal{P}\,\text{vec}(\tilde{\mathcal{I}}_F). \tag{27}$$

So, $\tilde{\mathcal{I}}_F$ can be defined as $\frac{1}{|D|} \sum_{i=1}^{|D|} (\mathbf{G}_i \otimes \mathbf{G}_i)$. This fact is used in the accelerated adaptation of the Kronecker Factorization algorithm.

Now, suppose a $\mathcal{I}_F$ and $\tilde{\mathcal{I}}_F$ are connected with $\mathcal{R} \in \mathbb{R}^{n \times n}$ (see Eq. 22):

$$\text{vec}(\widetilde{\mathcal{I}}_F) = (I \otimes \mathcal{R}) \cdot \text{vec}(\mathcal{I}_F), \mathcal{P} = I \otimes \mathcal{R} \tag{28}$$

## C  APPENDIX: RIGHT VECTOR-MATRIX MULTIPLICATION

We can define right vector-matrix multiplication as follows:

$$\mathcal{I}_F^\top z = (\sum_{i=1}^{|D|} \mathbf{G}_i \otimes \mathbf{G}_i)^\top z \tag{29}$$

Using property of the Kronecker product $(\mathbf{K} \otimes \mathbf{L})\,\text{vec}(\mathbf{C}) = \text{vec}(\mathbf{K}^\top \mathbf{C}\mathbf{L})$:

$$\mathcal{I}_F^\top z = \sum_{i=1}^{|D|} \text{vec}(\mathbf{G}_i \mathbf{Z} \mathbf{G}_i^\top), \text{ where } z = \text{vec}(\mathbf{Z}), \mathbf{Z} \in \mathbb{R}^{m \times m} \tag{30}$$

## D  APPENDIX: EXTENDED DECODER EVALUATION ON MMLU

Table 6 and Figure 3 shows that for LLaMA 2 7B GFWSVD consistently outperforms both simple and strong baselines across all compression rates. In particular, at the most aggressive compression setups (15–20% of the original parameters), our method matches or exceeds the accuracy of activation-based methods and shows substantially lower perplexities on both WikiText-2 and PTB.

We also compressed the Llama 3.1 8B model using ours GFWSVD and compared it to the activation-aware SVD (ASVD) method. Due to its extensive training on 15 trillion tokens, Llama 3.1 has exceptionally high information density and low parameter redundancy, therefore, it is a significantly more challenging target for compression than Llama 2. In Table 7 we show that Llama 3.1 has a stronger degradation in quality upon compression than Llama 2. Nevertheless, our method GFWSVD demonstrated better results across all compression ratios.

## E  APPENDIX: THROUGHPUT AND FLOPS FOR COMPRESSED MODELS

GFWSVD, ASVD, SVD-LLM compresses weight $\mathbf{W} \in \mathbb{R}^{n \times m}$ into a pair of low-rank matrices $\mathbf{W}_1 \in \mathbb{R}^{n \times r}$ and $\mathbf{W}_2 \in \mathbb{R}^{r \times m}$. This reduces the number of FLOPs required during the forward pass through a linear layer from $O(nm)$ to $O(nr + rm) = O(r(n + m))$.

We ran inference-time latency measurements on the Llama 2 7B model under different compressions. The results are shown below (averaged over 100 runs, batch size = 1, sequence length = 1024 tokens, GPU: A100 80GB).

Table 6: Performance of the Llama 2 7B Chat compressed by various methods under compression ratios from 5% to 20% on WikiText-2, PTB, and MMLU. Lower is better for perplexity (↓), higher is better for accuracy (↑). We denote compression ratio as $1 - \frac{\text{compressed model}}{\text{original model}}$.

| METHOD | WikiText-2↓ | PTB↓ | C. Ratio | MMLU Avg↑ | Humanities↑ | Other↑ | Social Sciences↑ | STEM↑ |
|---|---|---|---|---|---|---|---|---|
| Full model | **6.94** | **25.75** | 0% | 0.46 ± 0.003 | 0.43 ± 0.01 | 0.55 ± 0.01 | 0.53 ± 0.01 | 0.36 ± 0.01 |
| FWSVD (Hsu et al., 2022) | 7.52 | 45.25 | | 0.40 ± 0.003 | 0.36 ± 0.01 | 0.45 ± 0.01 | 0.45 ± 0.01 | 0.35 ± 0.01 |
| ASVD (Yuan et al., 2023) | 7.60 | **26.29** | 5% | **0.41** ± 0.004 | 0.37 ± 0.01 | **0.48** ± 0.01 | **0.46** ± 0.01 | **0.35** ± 0.01 |
| SVD-LLM (Wang et al., 2025c) | 8.80 | 51.28 | | 0.34 ± 0.004 | 0.31 ± 0.01 | 0.38 ± 0.01 | 0.35 ± 0.01 | 0.31 ± 0.01 |
| GFWSVD (Ours) | **7.16** | 28.55 | | 0.40 ± 0.003 | **0.38** ± 0.01 | 0.47 ± 0.01 | 0.44 ± 0.01 | 0.33 ± 0.01 |
| FWSVD (Hsu et al., 2022) | 11.53 | 96.62 | | 0.37 ± 0.004 | 0.34 ± 0.01 | 0.43 ± 0.01 | 0.42 ± 0.01 | 0.33 ± 0.01 |
| ASVD (Yuan et al., 2023) | 8.97 | 40.12 | 10% | 0.37 ± 0.004 | 0.33 ± 0.01 | 0.42 ± 0.01 | 0.40 ± 0.01 | 0.33 ± 0.01 |
| SVD-LLM (Wang et al., 2025c) | 9.69 | 60.82 | | 0.32 ± 0.004 | 0.30 ± 0.01 | 0.35 ± 0.01 | 0.32 ± 0.01 | 0.30 ± 0.01 |
| GFWSVD (Ours) | **8.77** | **36.44** | | **0.38** ± 0.002 | **0.35** ± 0.01 | **0.44** ± 0.01 | **0.42** ± 0.01 | **0.33** ± 0.01 |
| FWSVD (Hsu et al., 2022) | 22.06 | 411.50 | | 0.31 ± 0.009 | 0.29 ± 0.01 | 0.34 ± 0.01 | 0.33 ± 0.01 | 0.30 ± 0.01 |
| ASVD (Yuan et al., 2023) | 10.91 | 83.49 | 15% | 0.32 ± 0.003 | 0.30 ± 0.01 | 0.33 ± 0.01 | 0.32 ± 0.01 | 0.30 ± 0.01 |
| SVD-LLM (Wang et al., 2025c) | 10.36 | 72.58 | | 0.30 ± 0.004 | 0.29 ± 0.01 | 0.34 ± 0.01 | 0.31 ± 0.01 | 0.30 ± 0.01 |
| GFWSVD (Ours) | **10.06** | **42.19** | | **0.36** ± 0.004 | **0.33** ± 0.01 | **0.41** ± 0.01 | **0.38** ± 0.01 | **0.32** ± 0.01 |
| FWSVD (Hsu et al., 2022) | 66.37 | 1523.00 | | 0.27 ± 0.004 | 0.25 ± 0.01 | 0.30 ± 0.01 | 0.28 ± 0.01 | 0.28 ± 0.01 |
| ASVD (Yuan et al., 2023) | 27.73 | 241.57 | 20% | 0.26 ± 0.004 | 0.25 ± 0.01 | 0.27 ± 0.01 | 0.24 ± 0.01 | 0.28 ± 0.01 |
| SVD-LLM (Wang et al., 2025c) | 11.23 | 98.91 | | 0.29 ± 0.004 | 0.27 ± 0.01 | 0.32 ± 0.01 | 0.29 ± 0.01 | 0.29 ± 0.01 |
| GFWSVD (Ours) | **11.13** | **50.50** | | **0.32** ± 0.003 | **0.30** ± 0.01 | **0.35** ± 0.01 | **0.34** ± 0.01 | **0.30** ± 0.01 |

Table 7: Performance of Llama 3.1 8B Instruct compressed by various methods under compression ratios from 10% to 20% on WikiText-2, PTB, and MMLU. Lower is better for perplexity (↓), higher is better for accuracy (↑).

| METHOD | WikiText-2↓ | PTB↓ | Compr. | MMLU Avg↑ | Humanities↑ | Other↑ | Social Sciences↑ | STEM↑ |
|---|---|---|---|---|---|---|---|---|
| Full model | **7.2** | **11.50** | 0% | 0.68 ± 0.006 | 0.64 ± 0.01 | 0.73 ± 0.01 | 0.78 ± 0.01 | 0.60 ± 0.01 |
| ASVD (Yuan et al., 2023) | 10.91 | **19.33** | 10% | 0.39 ± 0.004 | 0.39 ± 0.01 | 0.33 ± 0.01 | 0.35 ± 0.01 | 0.35 ± 0.01 |
| GFWSVD (Ours) | **9.38** | 19.81 | | **0.54** ± 0.002 | **0.49** ± 0.01 | **0.62** ± 0.01 | **0.63** ± 0.01 | **0.46** ± 0.01 |
| ASVD (Yuan et al., 2023) | 38.02 | 76.1 | 15% | 0.29 ± 0.004 | 0.31 ± 0.01 | 0.32 ± 0.01 | 0.31 ± 0.01 | 0.31 ± 0.01 |
| GFWSVD (Ours) | **16.75** | **23.67** | | **0.50** ± 0.001 | **0.46** ± 0.01 | **0.56** ± 0.01 | **0.57** ± 0.01 | **0.43** ± 0.01 |
| ASVD (Yuan et al., 2023) | 145 | 1672 | 20% | 0.24 ± 0.003 | 0.27 ± 0.01 | 0.28 ± 0.01 | 0.27 ± 0.01 | 0.27 ± 0.01 |
| GFWSVD (Ours) | **22.57** | **32.4** | | **0.43** ± 0.003 | **0.39** ± 0.01 | **0.49** ± 0.01 | **0.48** ± 0.01 | **0.38** ± 0.01 |

Table 8: Comparison of theoretical FLOPs for Llama 2 7B Chat under different compression rates. All values are in trillions (T) of FLOPs.

| Model | Compression Ratio | Full Model FLOPs | Compressed FLOPs |
|---|---|---|---|
| Llama 2 7B | 10% | 53.05T | 42.43T |
| | 15% | 53.05T | 39.24T |
| | 20% | 53.05T | 37.18T |
| | 40% | 53.05T | 31.83T |
| | 50% | 53.05T | 29.37T |

Table 9: Throughput (tokens/s) achieved by the uncompressed Llama 2 7B Chat and its FWSVD-compressed versions (batch size = 1, sequence length = 1024).

| Compression Ratio | Tokens/s | Relative Speedup |
|---|---|---|
| 0% (Uncompressed) | 1186 | 1.00× |
| 10% | 1269 | 1.07× |
| 15% | 1294 | 1.09× |
| 20% | 1323 | 1.12× |
| 40% | 1510 | 1.27× |
| 50% | 1600 | 1.34× |

We ran inference-time latency measurements on the Llama 7B model using different compression ranks. The results are shown in Table 10(averaged over 100 runs, batch size = 1, sequence length = 1024 tokens, GPU: A100 80GB).

Table 10: Inference latency (in milliseconds) per token for compressed Llama 2 7B Chat model. Lower is better. Reported values represent forward pass time averaged over 100 runs.

| Method | Compression | | | | | |
|---|---|---|---|---|---|---|
| | 0% (Uncompressed) | 10% | 15% | 20% | 40% | 50% |
| Throughput↓ (`batch_size=64`) | | | | | | |
| GFWSVD (Ours) | 4.7 | 4.5 | 4.2 | 4.0 | 3.3 | 2.95 |
| SVD-LLM | 4.7 | 4.4 | 4.2 | 3.9 | – | – |
| Throughput↓ (`batch_size=16`) | | | | | | |
| GFWSVD (Ours) | 3.2 | 3.0 | 2.8 | 2.6 | 2.3 | 2.15 |
| SVD-LLM | 3.2 | 2.9 | 2.8 | 2.6 | – | |

# F  APPENDIX: EXTENDED GLUE RESULTS

We report extended compression results on tasks of GLUE benchmark in Table 11.

Table 11: Performance of BERT model compressed by various methods under compression rates from 60% to 99% on GLUE benchmark. Higher is better for all tasks (↑).

| METHOD / DATASET | MRPC↑ | STSB↑ | QQP↑ | MNLI↑ | QNLI↑ | RTE↑ | COLA↓ | SST2↑ |
|---|---|---|---|---|---|---|---|---|
| Full model | **0.77** | **0.87** | **0.90** | **0.83** | **0.90** | **0.56** | **0.41** | **0.91** |
| | | | | Compression Ratio 1% ($r = 600$) | | | | |
| SVD | 0.67 | 0.84 | 0.90 | 0.67 | 0.90 | 0.56 | 0.58 | 0.91 |
| ASVD (Yuan et al., 2023) | 0.72 | 0.73 | 0.89 | 0.83 | 0.90 | 0.56 | 0.41 | 0.91 |
| FWSVD (Hsu et al., 2022) | 0.72 | 0.87 | 0.90 | 0.72 | 0.90 | 0.55 | 0.36 | 0.91 |
| GFWSVD (Ours) | **0.73** | **0.87** | **0.90** | **0.73** | **0.90** | **0.56** | **0.55** | **0.92** |
| | | | | Compression Ratio 8% ($r = 500$) | | | | |
| SVD | 0.53 | 0.82 | 0.89 | 0.53 | 0.90 | 0.54 | 0.53 | 0.89 |
| ASVD (Yuan et al., 2023) | 0.71 | 0.56 | 0.86 | 0.81 | 0.89 | 0.53 | 0.44 | 0.88 |
| FWSVD (Hsu et al., 2022) | 0.71 | 0.87 | 0.90 | 0.71 | 0.89 | 0.56 | 0.34 | 0.91 |
| GFWSVD (Ours) | **0.73** | **0.87** | **0.90** | **0.73** | **0.90** | **0.56** | **0.49** | **0.92** |
| | | | | Compression Ratio 23% ($r = 250$) | | | | |
| SVD | 0.49 | 0.68 | 0.81 | 0.49 | 0.85 | 0.50 | 0.17 | 0.57 |
| ASVD (Yuan et al., 2023) | 0.69 | 0.08 | 0.76 | 0.50 | 0.58 | 0.47 | 0.11 | 0.75 |
| FWSVD (Hsu et al., 2022) | 0.69 | 0.86 | 0.89 | 0.69 | 0.89 | 0.61 | 0.23 | 0.80 |
| GFWSVD (Ours) | **0.71** | **0.86** | **0.89** | **0.71** | **0.89** | **0.61** | **0.38** | **0.88** |
| | | | | Compression Ratio 33% ($r = 100$) | | | | |
| SVD | 0.32 | 0.08 | 0.64 | 0.32 | 0.80 | 0.51 | 0.01 | 0.49 |
| ASVD (Yuan et al., 2023) | 0.58 | 0.07 | 0.74 | 0.39 | 0.50 | 0.47 | 0.05 | 0.82 |
| FWSVD (Hsu et al., 2022) | 0.69 | 0.58 | 0.87 | 0.71 | 0.86 | 0.55 | 0.21 | 0.72 |
| GFWSVD (Ours) | **0.71** | **0.70** | **0.87** | **0.71** | **0.86** | **0.55** | **0.21** | **0.72** |
| | | | | Compression Ratio 36% ($r = 50$) | | | | |
| SVD | 0.32 | 0.19 | 0.57 | 0.32 | 0.78 | 0.48 | 0.02 | 0.49 |
| ASVD (Yuan et al., 2023) | 0.68 | 0.03 | 0.73 | 0.49 | 0.76 | 0.51 | 0.03 | 0.80 |
| FWSVD (Hsu et al., 2022) | 0.69 | 0.65 | 0.84 | 0.69 | 0.72 | 0.46 | 0.03 | 0.77 |
| GFWSVD (Ours) | **0.69** | **0.65** | **0.84** | **0.69** | **0.72** | **0.46** | **0.05** | **0.77** |
| | | | | Compression Ratio 39% ($r = 10$) | | | | |
| SVD | 0.32 | 0.32 | 0.67 | 0.32 | 0.61 | 0.51 | 0.00 | 0.49 |
| ASVD (Yuan et al., 2023) | 0.61 | 0.14 | 0.64 | 0.40 | 0.57 | 0.49 | -0.04 | 0.76 |
| FWSVD (Hsu et al., 2022) | 0.37 | 0.32 | 0.79 | 0.37 | 0.57 | 0.49 | 0.00 | 0.49 |
| GFWSVD (Ours) | **0.53** | **0.60** | **0.79** | **0.53** | **0.62** | **0.47** | **0.05** | **0.65** |
| | | | | Compression Ratio 40% ($r = 1$) | | | | |
| SVD | 0.32 | 0.04 | 0.69 | 0.31 | 0.55 | 0.53 | 0.00 | 0.49 |
| ASVD (Yuan et al., 2023) | 0.62 | 0.10 | 0.64 | 0.42 | 0.50 | 0.49 | 0.03 | 0.70 |
| FWSVD (Hsu et al., 2022) | 0.32 | 0.18 | 0.72 | 0.32 | 0.51 | 0.50 | 0.00 | 0.49 |
| GFWSVD (Ours) | **0.42** | **0.70** | **0.74** | **0.42** | **0.65** | **0.52** | **0.05** | **0.49** |

# G  APPENDIX: IMPACT OF DIAGONAL AND NON-DIAGONAL ELEMENTS OF FACTORS

To assess the significance of diagonal elements, we performed the following ablation study. In the resulting factor matrices we retained either (1) only the off-diagonal elements (**Non-diag**) or (2) only the diagonal elements (**Diag**), and measured perplexity relative to our method and FWSVD. The results are in Table 12: the **Diag** variant performs better than FWSVD but worse than GFWSVD. This is expected, since FWSVD captures importance only along rows (only the left factor matrix has

a non-identity diagonal, see Fig. 1), whereas Non-diag GFWSVD captures both row and column importance. The contribution of off-diagonal elements provides a noticeable improvement compared to FWSVD.

Table 12: Perplexity ($\downarrow$) at 90% and 85% compression rates for GFWSVD with full, diagonal-only and non-diagonal factors for LLaMA 2 7B Chat compression.

| METHOD / DATASET | WikiText 2$\downarrow$ | PTB$\downarrow$ | WikiText 2$\downarrow$ | PTB$\downarrow$ |
|---|---|---|---|---|
| Compression | 10% | 10% | 15% | 15% |
| FWSVD (Hsu et al., 2022) | 11.53 | 96.62 | 22.00 | 411.00 |
| Diag GFWSVD | 10.94 | 45.26 | 11.06 | 48.25 |
| Non-diag GFWSVD | 8.85 | 37.25 | 10.22 | 43.75 |
| Full GFWSVD (Ours) | **8.77** | **36.44** | **10.06** | **42.19** |

## H  APPENDIX: LIMITATIONS

Our method decomposes the observed Fisher information matrix $\mathcal{I}_F$ into a Kronecker product of two smaller matrices, $\mathbf{Y}$ and $\mathbf{X}$ (Eq. 12). While effective, this assumes exact factorization, which may not hold in practice and can limit approximation quality and task sensitivity. In LLM experiments, we also observed cases where the estimated Kronecker factors were singular, requiring regularization (e.g., $\mathbf{Y} \leftarrow \mathbf{Y} + \alpha \operatorname{diag} \mathbf{Y}$) to ensure positive definiteness and numerical stability. Although this resolves instability, it introduces additional computational overhead.

We observed that compression effectiveness varies significantly across layers, making preliminary layer selection necessary to achieve favorable trade-offs. A key limitation of our current approach is the lack of coordination across layers during compression. For effective multi – layer compression—especially in large-scale models like LLMs – it is important to account for cross-layer dependencies. Future work could focus on modeling these interactions to enable joint compression strategies.

## I  APPENDIX: LLM USAGE STATEMENT

We used large language models (LLMs) only as a general-purpose writing assistant for grammar checking and text polishing. The research ideas, implementation, analysis, and conclusions are entirely our own.

