# OpenReview forum: "Generalized Fisher-Weighted SVD: Scalable Kronecker-Factored Fisher Approximation for Compressing Large Language Models."
_ICLR.cc/2026/Conference — Submitted to ICLR 2026_

### Official Review · Reviewer_CCmb · 2025-10-28

**Soundness:** 3
**Presentation:** 4
**Contribution:** 2
**Rating:** 4
**Confidence:** 3

**Summary:**

This paper introduces Generalized Fisher-Weighted SVD (GFWSVD), a post-training compression method for LLMs. It addresses the limitations of prior work like FWSVD, which uses only diagonal approximations of the Fisher Information Matrix (FIM) and ignores parameter correlations. GFWSVD leverages a Kronecker-factored approximation ($A \otimes B$) of the FIM to capture both row and column dependencies, integrating these factors into a generalized SVD framework. The authors provide a scalable algorithm to compute these factors efficiently and demonstrate empirically that GFWSVD outperforms existing gradient- and activation-based compression baselines (FWSVD, ASVD, SVD-LLM) on BERT and LLaMA-2 models.

**Strengths:**

1. The paper is well-written and clearly structured.
2. The proposed method GFWSVD is novel and  backed with theoretical grounding and practical evaluations, effectively generalizing FWSVD.

**Weaknesses:**

1. Evaluations on modern LLMs are lacking. Modern LLMs like Llama 3.1/Qwen 3 are known to be overtrained and harder to compress. It is important to provide evaluations on such models for the proposed method to become practical.
2. The paper can benefit from a more detailed literature review. More recent SVD-based LLM compression approaches can achieve much higher compression ratios on more recent LLMs; for example, BitStack[1] matches quantization results and far surpasses the baselines in the paper. I understand this paper is more analytical, but a section with a more detailed discussion of the practicality of GFWSVD and more recent approaches is needed.

[1] BitStack: Any-Size Compression of Large Language Models in Variable Memory Environments

**Questions:**

See Weaknesses. I’ll increase the score from 4 to 6/8 if the authors can provide the requested experiments and discussions.

---

> ### Author Response · Authors · 2025-11-21
>
> Thank you for your thoughtful review. Below, we outline the comments and the corresponding revisions.
>
> > Evaluations on modern LLMs are lacking.
>
> We compressed $\texttt{Llama 3.1 8B Instruct}$ (results are added in the Section 5.2) and compared the results with an ASVD. As expected, $\texttt{Llama 3.1 8B Instruct}$ quickly loses the performance under compression than $\texttt{Llama 2 7B Chat}$ due to its high information density. However, GFWSVD consistently outperforms the ASVD baseline. We duplicate the results below:
>
> | C_ratio            | Compression | WikiText-2 ↓ | PTB ↓  | MMLU Avg ↑ |
> |-------------------|:-----------:|:------------:|:------:|:-----------:|
> | **Full model**    |   **0%**  |   **7.2**    | **11.50** | **0.68** |
> |                   |             |              |          |            |
> | ASVD              |     10%     |    10.91     | **19.33** |    0.39    |
> | **GFWSVD (Ours)** |     10%     |  **9.38**    |  19.81   |  **0.54**  |
> |                   |             |              |          |            |
> | ASVD              |     15%     |    38.02     |   76.1   |    0.29    |
> | **GFWSVD (Ours)** |     15%     | **16.75**    | **23.67**|  **0.50**  |
> |                   |             |              |          |            |
> | ASVD              |     20%     |    1800      |  4234    |    0.24    |
> | **GFWSVD (Ours)** |     20%     | **22.57**    | **32.4** |  **0.43**  |
>
> > More recent SVD-based LLM compression approaches can achieve much higher compression ratios
>
> > The paper can benefit from a more detailed literature review.
>
> Most SOTA post-training compression methods use additional stochastic optimization and combine multiple techniques. For example, BitStack[1] uses post-quantization fine-tuning, BLAST[2] relies on gradient-descent factorization, and Dobi-SVD[3] optimizes singular values and then applies quantization.
>
> GFWSVD is different: it is fully analytical, with no stochastic steps. It is essentially a standard SVD, but re-weighted in a way that it becomes provably optimal for the given LLM. This makes our method orthogonal and complementary to pipeline-based approaches. For this reason, our experiments compared GFWSVD with methods of the same class — simple analytical approaches without extra optimization, which typically do not achieve high compression rates.
>
> To clarify the positioning of our method, we added discussions on BitStack [1], Dobi-SVD [3] and BLAST [2] in the Related Work section, noting that these approaches are pipelines that rely on stochastic optimization.
>
> We also added a new Section 5.3 ”Positioning and applicability” where we explain generalizability of our GFWSVD. We integrate GFWSVD into the Dobi-SVD pipeline and achieve 40\% compression with strong performance, exceeding the baseline that uses standard SVD.
>
> We describe the results of these experiments in Table 5 of the paper and duplicate it below for your convenience:
>
> | Method | C_ratio | WikiText-2 $\downarrow$ | PTB $\downarrow$ | ARC-E $\uparrow$ | ARC-C $\uparrow$ | PIQA $\uparrow$ | HellaSwag $\uparrow$ |
> |---|---|---|---|---|---|---|---|
> | **Full model** | **0%** | **6.94** | **25.75** | **0.73** | **0.44** | **0.78** | **0.57** |
> |  |  |  |  |  |  |  |  |
> | Dobi-SVD [1] | 20% | 7.75 | 26.11 | 0.71 | 0.40 | 0.76 | 0.55 |
> | **Dobi-GFWSVD (Ours)** | 20% | **7.56** | **25.95** | **0.72** | **0.42** | **0.77** | 0.55 |
> | *YAQA (quantization)* | 20% | *6.99* | -- | *0.73* | *0.44* | *0.78* | *0.56* |
> |  |  |  |  |  |  |  |  |
> | Dobi-SVD [1] | 40% | 10.56 | 41.70 | 0.58 | 0.32 | 0.69 | 0.32 |
> | **Dobi-GFWSVD (Ours)** | 40% | **10.29** | **38.56** | **0.60** | **0.33** | 0.69 | 0.32 |
> | *YAQA (quantization)* | 40% | *8.14* | -- | *0.66* | *0.41* | 0.77 | *0.53* |
>
> #### References
>
> [1] BitStack: Any-Size Compression of Large Language Models in Variable Memory Environments, 2024
>
> [2] BLAST: Block-Level Adaptive Structured Matrices for Efficient Deep Neural Network Inference, 2024
>
> [3] Dobi-SVD: Differentiable SVD for LLM Compression and Some New Perspectives, 2025

---

> > ### Comment · Reviewer_CCmb · 2025-11-27
> >
> > Thank you for the response. Regarding the following discussion:
> > > Most SOTA post-training compression methods use additional stochastic optimization and combine multiple techniques. For example, BitStack[1] uses post-quantization fine-tuning, BLAST[2] relies on gradient-descent factorization, and Dobi-SVD[3] optimizes singular values and then applies quantization.
> >
> > To the best of my knowledge, although the descriptions of BLAST and Dobi-SVD are correct, there is a factual error regarding BitStack. It uses neither quantization nor fine-tuning; instead, it is actually training-free and relies solely on SVD, which falls into the same category as GFWSVD. Can you clarify why this method is not compared?

---

> ### Author Response · Authors · 2025-12-02
> **More clarification on BitStack**
>
> You are right and we apologize for the factual inaccuracy in our previous comment (now corrected in the paper). BitStack is indeed a training-free approach. However, we treat BitStack as a different category of compression methods and will elaborate below.
>
> This is a memory management method, which iteratively decomposes each weight matrix according to parameter importance, producing residual blocks of roughly one bit per parameter. These blocks are stored in sequence and can be flexibly loaded at inference time depending on the available memory.
>
> BitStack relies on Iterative Absolute Value Decomposition, which splits weights into a sign matrix and a magnitude matrix. The sign matrix remains at full rank, therefore, the "skeleton" of the model remains the same, and we do not observe any significant performance decrease.
>
> Our GFWSVD is a pure decomposition method. GFWSVD physically removes parameters, which is a fundamental limitation, BitStack iteratively builds the precision of the number by adding 1-bit residual blocks.
>
> Taking a closer look at the BitStack reference implementation as its efficiency relies on custom Triton kernels to unpack 1-bit during inference time. In contrast, our GFWSVD operates entirely with standard matrices with floating point numbers. Thus, it is compatible with any off-the-shelf hardware or BLAS library.
>
> Therefore, we believe that BitStack remains outside the scope of our work. This is also evidenced by our experiments.
> We extended our setup to 50% compression and added a new baseline, Basis Sharing, which accounts for cross-layer correlations. We also evaluated BitStack at the same compression ratios on $\texttt{Llama 2 7B Chat}$ and $\texttt{Llama 3.1 8B Instruct}$. At 50% compression, both GFWSVD and Basis Sharing collapse: on ARC-C and OpenBookQA the scores are close to random. BitStack shows either no performance decrease or only about a 1%-4% decrease.
>
> The original BitStack paper reports the same pattern: Table 1 compares BitStack mostly to quantization methods, and Appendix A6 shows that BitStack clearly outperforms all decomposition-based methods.
>
> ## $\texttt{Llama 3.1 8B Instruct}$ — Metrics
>
> Here **c_rate** is 1 - $\frac{\text{compressed model}}{\text{orig model}}$
>
> | Method        | WikiText 2 | c_rate | ARC-C | ARC-E | HellaSwag | PIQA | WinoGrande | OpenBook | AVG  |
> |------------|---------------|----------|---------|--------|--------|------------|-------|-------------|-----------|
> | Full model    | 7.20     | 0%     | 0.52   | 0.81   | 0.59       | 0.79  | 0.73        | 0.35      | 0.63  |
> |------------|---------------|----------|---------|--------|--------|------------|-------|-------------|-----------|
> | ASVD          | 1456.00  | 20%    | 0.21   | 0.33   | 0.27       | 0.61  | 0.54        | 0.15      | 0.35  |
> | BASIS SHAR    | 18.54    | 20%    | 0.34   | 0.68   | 0.42       | 0.70  | 0.65        | 0.35      | 0.52  |
> | GFWSVD        | 22.57    | 20%    | 0.35   | 0.68   | 0.45       | 0.75  | 0.63        | 0.33      | 0.53  |
> | BitStack      | 8.60     | 20%    | 0.51   | 0.81   | 0.58       | 0.79  | 0.73        | 0.34      | 0.63  |
> |------------|---------------|----------|---------|--------|--------|------------|-------|-------------|-----------|
> | ASVD          | 145.00   | 30%    | 0.22   | 0.30   | 0.25       | 0.58  | 0.52        | 0.16      | 0.34  |
> | BASIS SHAR    | 32.71    | 30%    | 0.29   | 0.52   | 0.43       | 0.63  | 0.60        | 0.31      | 0.46  |
> | GFWSVD        | 35.93    | 30%    | 0.33   | 0.61   | 0.42       | 0.71  | 0.58        | 0.23      | 0.48  |
> | BitStack      | 7.49     | 30%    | 0.50   | 0.81   | 0.58       | 0.79  | 0.73        | 0.34      | 0.63  |
> |------------|---------------|----------|---------|--------|--------|------------|-------|-------------|-----------|
> | ASVD          | 2992.00  | 40%    | 0.23   | 0.27   | 0.26       | 0.55  | 0.49        | 0.15      | 0.33  |
> | BASIS SHAR    | 78.52    | 40%    | 0.24   | 0.39   | 0.33       | 0.55  | 0.56        | 0.28      | 0.39  |
> | GFWSVD        | 69.56    | 40%    | 0.25   | 0.41   | 0.32       | 0.61  | 0.55        | 0.22      | 0.39  |
> | BitStack      | 7.98     | 40%    | 0.51   | 0.81   | 0.58       | 0.79  | 0.73        | 0.34      | 0.63  |
> |------------|---------------|----------|---------|--------|--------|------------|-------|-------------|-----------|
> | ASVD          | 40939.00 | 50%    | 0.22   | 0.26   | 0.26       | 0.50  | 0.48        | 0.13      | 0.31  |
> | BASIS SHAR    | 203.43   | 50%    | 0.23   | 0.30   | 0.29       | 0.52  | 0.53        | 0.27      | 0.35  |
> | GFWSVD        | 176.68   | 50%    | 0.23   | 0.31   | 0.28       | 0.55  | 0.54        | 0.22      | 0.36  |
> | BitStack      | 8.77     | 50%    | 0.51   | 0.81   | 0.57       | 0.79  | 0.72        | 0.33      | 0.62  |

---

> ### Author Response · Authors · 2025-12-02
>
> ## $\texttt{Llama 2 7B Chat}$ — Metrics
>
> | Method        | c_rate | ARC-C | ARC-E | HellaSwag | PIQA | WinoGrande | OpenBook | AVG  |
> |---------------|----------|--------|--------|--------|------------|-------|-------------|-----------|
> | Full model    | 0%     | 0.44   | 0.73   | 0.58       | 0.76  | 0.67        | 0.33      | 0.59  |
> |---------------|----------|--------|--------|--------|------------|-------|-------------|-----------|
> | ASVD          | 20%    | 0.27   | 0.51   | 0.39       | 0.68  | 0.61        | 0.22      | 0.45  |
> | SVD-LLM       | 20%    | 0.29   | 0.66   | 0.40       | 0.66  | 0.61        | 0.23      | 0.48  |
> | BASIS SHARING |  20%    | 0.31   | 0.65   | 0.42       | 0.68  | 0.61        | 0.27      | 0.49 |
> | GFWSVD        |  20%    | 0.33   | 0.62   | 0.47       | 0.74  | 0.61        | 0.25      | 0.50  |
> | BitStack      |20%    | 0.43   | 0.73   | 0.57       | 0.76  | 0.66        | 0.31      | 0.58  |
> |---------------|----------|--------|--------|--------|------------|-------|-------------|-----------|
> | ASVD          | 30%    | 0.21   | 0.31   | 0.29       | 0.63  | 0.54        | 0.15      | 0.36  |
> | SVD-LLM       | 30%    | 0.25   | 0.52   | 0.34       | 0.62  | 0.55        | 0.22      | 0.42  |
> | BASIS SHARING | 30%    | 0.27   | 0.58   | 0.38       | 0.63  | 0.58        | 0.26      | 0.45 |
> | GFWSVD        | 30%    | 0.28   | 0.56   | 0.40       | 0.63  | 0.58        | 0.20      | 0.44  |
> | BitStack      |  30%    | 0.42   | 0.72   | 0.56       | 0.75  | 0.66        | 0.30      | 0.57 |
> |---------------|--------|--------|--------|------------|-------|-------------|-----------|-------|
> | ASVD          | 40%    | 0.22   | 0.26   | 0.26       | 0.49  | 0.49        | 0.16      | 0.31  |
> | SVD-LLM       |  40%    | 0.26   | 0.45   | 0.30       | 0.55  | 0.54        | 0.19      | 0.38  |
> | BASIS SHARING | 40%    | 0.21   | 0.46   | 0.32       | 0.58  | 0.55        | 0.19      | 0.39 |
> | GFWSVD        | 40%    | 0.27   | 0.48   | 0.33       | 0.64  | 0.57        | 0.17      | 0.41  |
> | BitStack      |  40%    | 0.41   | 0.72   | 0.56       | 0.76  | 0.67        | 0.31      | 0.57 |
> |---------------|--------|--------|--------|------------|-------|-------------|-----------|-------|
> | ASVD          |50%    | 0.21   | 0.25   | 0.26       | 0.53  | 0.49        | 0.16      | 0.32  |
> | SVD-LLM       |  50%    | 0.21   | 0.33   | 0.26       | 0.54  | 0.50        | 0.12      | 0.33  |
> | BASIS SHARING |  50%    | 0.20   | 0.36   | 0.30       | 0.55  | 0.50        | 0.15      | 0.34 |
> | GFWSVD        | 50%    | 0.22   | 0.28   | 0.26       | 0.55  | 0.51        | 0.15      | 0.33  |
> | BitStack      | 50%    | 0.41   | 0.71   | 0.56       | 0.75  | 0.67        | 0.30      | 0.57 |

---

### Official Review · Reviewer_kykp · 2025-10-29

**Soundness:** 2
**Presentation:** 2
**Contribution:** 2
**Rating:** 2
**Confidence:** 4

**Summary:**

This work proposes Generalized Fisher-Weighted SVD (GFWSVD), a post-training LLM compression technique. The method uses the Fisher information to approximate the Hessian, which is then further approximated using a Kronecker product.
Since leveraging the full Fisher information is computationally expensive for large-scale models, existing approaches typically rely on diagonal approximations. This work tightens that approximation by proposing a more general method. The authors develop theory to motivate their approach, assuming the model's dense weight matrices are drawn from a matrix-variate normal distribution. Empirically, they demonstrate that their method achieves better perplexity and accuracy compared to (some) existing methods.

**Strengths:**

The strength of this paper is that it generally seems to outperform the baselines in terms of perplexity and accuracy. This can be observed in Tables 3 and 4 of the main paper, along with Table 8 in the appendix. It also tightens the diagonal approximation of the Fisher information used in other works, albeit under the assumption that the weight matrices are drawn from a matrix-variate normal distribution.

**Weaknesses:**

I think one of the major weaknesses of this paper is the clarity in presentation, in that it is often missing key details to either experiments or is not clear to understand. For example:
- In Theorem 1, what is the "task loss function"? I don't think the "task" is clearly defined. I assume that the authors are referring to the problem in Equation (5), but then this should be referenced.  In fact, is this first condition (or perhaps the second condition) redundant? Based on lines 154-156 of the manuscript, it seems to me that 1 and 3 already imply 2, or 2 and 3 already imply 1. I think the authors can frame this a bit more clearly.
- Table 1 is difficult to follow. Is this the time it takes to perform the decomposition of all of the matrices in the corresponding model (e.g., LLaMa-2-7B) or is it just a few matrices of the model? If it is a few, then which matrices are chosen? What are the dimensions of these matrices? Or is this the time taken to approximate the actual Fisher information matrix?
- What is the number of batches $D$ used to approximate the Fisher information in your experiments?
- Separating Tables 2 and 3 is a bit confusing to me; it seems to me that you can include Table 2 as a row in Table 3.

I'm also concerned at the compression rates used in the experiments; as far as I am aware, a 20% compression rate as done in Table 4 is pretty low and existing works compress at a far more aggressive rate and have good performance -- see [1] as an example.

---

[1] "BLAST: Block-Level Adaptive Structured Matrices for Efficient Deep Neural Network Inference". Changwoo Lee, Soo Min Kwon, Qing Qu, Hun-Seok Kim. NeurIPS 2024.

**Questions:**

I have listed a few questions in the weaknesses section. Here are just a few more:

- Why is assuming that the dense weight matrices follow a matrix-variate normal distribution a fair assumption? As far as I can tell, this generalized decomposition only holds under this assumption. Can you reference existing works that also make this assumption, or is there any evidence you can provide?
- In the limitations section, it says that often a constant $\alpha$ is used to ensure positive semidefiniteness. What is the constant used for experiments? Does FWSVD suffer from this issue as well?

I apologize in advance if these questions are already answered in the manuscript and I missed them. If so, I would appreciate it if the authors could kindly point me to the relevant sections.

---

> ### Author Response · Authors · 2025-11-21
>
> We are grateful for the review and detailed comments. We have addressed every point raised and believe the manuscript is substantially improved.
>
> **Weakness**
>
> > In Theorem 1, what is the "task loss function"?
>
> We thank the reviewer for their thoughtful question regarding the conditions in Theorem 1.
>
> To clarify the conditions:
>
> 1) The "task loss function" is the neural network's original training objective (e.g., cross-entropy for language modeling), which is a Maximum Likelihood Estimation (MLE) objective. This establishes the fundamental connection between the Hessian and the Fisher Information Matrix (FIM) at convergence, a well-established principle for such objectives.
>
> 2) Condition 2 (Kronecker structure) formalizes our core methodological contribution. While the general MLE property gives us $H \approx B$, the Kronecker structure $F \approx A \otimes B$ is a specific approximation we make to render the problem tractable for large models.
>
> 3) Condition 3 (MVN prior) provides the probabilistic motivation for the compression objective in Equation (8).
>
> It should be noted that Condition 2 does not generally hold exactly, as neural networks often exhibit complex, non-Kronecker Hessian structure. Therefore, we treat Condition 2 as an operative Hessian approximation that enables tractable computation.
>
> Thus, Theorem 1 relies on three conceptual components to derive the optimal compression. Condition 1 establishes the equivalence between the Hessian and the FIM in MLE problems. Condition 2 introduces the key structural approximation to the FIM, which is essential for computational scalability. Condition 3 specifies the probabilistic model that motivates the compression objective in Equation (8).
>
> We have added these clarifications to the revised version of the paper.
>
> > Table 1 is difficult to follow.
>
> Table 1 reports the average Hessian decomposition time for a single linear layer, not the whole model.
>
> In Section 4.1, we propose an accelerated Hessian decomposition algorithm. However, in the context of LLM, this algorithm is practical as long as a single layer can be decomposed quickly. This is the relevant bottleneck, since each layer has a very large Hessian (Lines L273–L276 in the previous paper version). Table 1 shows that this accelerated algorithm works well on many transformer models.
>
> We added more clarification to the text.
>
> > What is the number of batches used to approximate the Fisher information?
>
> For LLMs compression we use a subset of FineWeb dataset: a calibration dataset with approximately 200K tokens (45 batches in our case). For BERT, we compute Fisher information using one full epoch of the downstream task data. The specific batch counts for GLUE tasks are below:
>
> | Dataset | Batch Size | $D$ |
> |:---|:---:|:---:|
> |**STSB**| 32 | 180 |
> |**CoLA**| 32 | 268 |
> |**MNLI**| 32 | 12 272 |
> |**MRPC**| 32 | 115|
> |**QNLI**| 16 | 6 547|
> |**QQP**| 32 | 11 371|
> |**RTE**| 16 | 156|
> |**SST2**| 32 | 2 105 |
>
> > Separating Tables 2 and 3 is a bit confusing
>
> Indeed, the tables can be merged without any loss of information, and we have done so in the revised version.
>
> > Concerning 20\% compression rate.
>
> Most SOTA post-training compression methods use additional stochastic optimization and combine multiple techniques. For example, BLAST[1] relies on gradient-descent factorization, and Dobi-SVD[2] optimizes singular values and then applies quantization (we added the description of the more advanced baselines in the Related Work section).
>
> GFWSVD is different: it is fully deterministic, with no stochastic steps. It is essentially a standard SVD, but re-weighted in a way that it becomes provably optimal for the given LLM. For this reason, our experiments compared GFWSVD with methods of the same class - simple analytical approaches without extra optimization, which typically do not achieve high compression rates.
>
> In the updated paper version, we integrate GFWSVD into the **Dobi-SVD** pipeline and achieve $40\%$ compression with strong performance, exceeding the baseline that uses standard SVD. Results are in the new **Table 5** in **Section 5.3**. They show that Dobi-GFWSVD beats Dobi-SVD at both $20\%$ and $40\%$ compression across all benchmarks. This shows our Fisher weighting provides a better starting point even when you do add training afterward.
>
> We also added experiments with $\texttt{Llama 3.1 8B Instruct}$ to the **Section 5.2, Table 4**. As expected, the model quickly loses the original performance under compression than Llama 2 due to its high information density. However, GFWSVD consistently outperforms the ASVD baseline. Most notably, at 20% reduction, ASVD fails completely (with PPL of $1800$ on WikiText 2 and $4234$ on PTB), whereas GFWSVD remains stable (with PPL of $22.57$ on WikiText 2 and $32.4$ on PTB).
>
> [1] BLAST:Block-Level Adaptive Structured Matrices for Efficient Deep Neural Network Inference, 2024
> [2] Dobi-SVD: Differentiable SVD for LLM Compression and Some New Perspectives

---

> ### Author Response · Authors · 2025-11-21
> **Reviewer’s questions**
>
> > About MVN assumption
>
> This assumption, although a simplification, is often considered a fair assumption because the Central Limit Theorem can be applied to the layer's weights. For example, in the work by Blundell et al. (2015) - "Weight Uncertainty in Neural Networks" and in many works on Bayesian Neural Networks, the prior distribution of weights is assumed to be normal.
>
> > About constant
>
> No, FWSVD does not suffer from this non–positive-definiteness issue because it does not require computing a matrix square root.
>
> To determine the optimal constant for a given layer, we iteratively increase $\alpha$, starting from a $10^6$ and multiplying it by 10 at each step until the resulting matrix becomes positive definite. This procedure reflects a trade-off between adding as little noise as possible while ensuring positive definiteness. On average, this process takes about 5 seconds, and for LLaMA-2-7B it is required for roughly 70% of the layers.

---

> ### Author Response · Authors · 2025-11-27
> **Additional experiments at 50% compression**
>
> We also added results for compressing $\texttt{Llama 2 7B Chat}$ and $\texttt{Llama 3.1 8B Instruct}$ down to 50% and evaluated them on ARC-C, ARC-E, HellaSwag, PIQA, WinoGrande, and OpenBook. We additionally included Basis Sharing [1] as a baseline — an SVD-pruning method that leverages inter-layer correlation information.
>
> We do not report SVD-LLM for LLaMA-3, since SVD-LLM does not provide a patch for $\texttt{Llama 3.1 8B Instruct}$.
>
> As can be seen, GFWSVD outperforms SVD-LLM across all compression ratios, and it surpasses Basis Sharing on $\texttt{Llama 2 7B Chat}$. However, for $\texttt{Llama 2 7B Chat}$ it lags behind Basis Sharing at 70% and 50% parameters retained. This is quite interesting and might be explained by the fact that GFWSVD utilizes information about parameter correlation within a layer, while Basis Sharing utilizes information about layer correlation within the model. For the "overfitted" $\texttt{Llama 3.1 8B Instruct}$, the information regarding weight correlation is of greater significance than it is for $\texttt{Llama 2 7B Chat}$.
>
> Here **c_ratio** is 1 - $\frac{\text{compressed model}}{\text{orig model}}$
>
> **$\texttt{Llama 3.1 8B Instruct}$**
>
> | C_ratio | Method        | ARC-C | ARC-E | HellaSwag | PIQA | WinoGrande | OpenBook | AVG  |
> |---------------------|---------------|-------|-------|-----------|------|------------|----------|------|
> | **100%**            | Full model    | 0.52  | 0.81  | 0.59      | 0.79 | 0.73       | 0.35     | 0.63 |
> |---------------------|---------------|-------|-------|-----------|------|------------|----------|------|
> | **20%**             | Basis Sharing | 0.34  | 0.68  | 0.42      | 0.70 | 0.65       | 0.35     | 0.52 |
> |                     | **GFWSVD**    | **0.35** | **0.68** | **0.45** | **0.75** | **0.63** | **0.33** | **0.53** |
> |---------------------|---------------|-------|-------|-----------|------|------------|----------|------|
> | **30%**             | Basis Sharing | 0.29  | 0.52  | 0.43      | 0.63 | 0.60       | 0.31     | 0.46 |
> |                     | **GFWSVD**    | **0.33** | **0.61** | **0.42** | **0.71** | **0.58** | **0.23** | **0.48** |
> |---------------------|---------------|-------|-------|-----------|------|------------|----------|------|
> | **40%**             | Basis Sharing | 0.24  | 0.39  | 0.33      | 0.56 | 0.56       | 0.28     | 0.39 |
> |                     | **GFWSVD**    | **0.23** | **0.41** | **0.31** | **0.61** | **0.54** | **0.22** | **0.39** |
> |---------------------|---------------|-------|-------|-----------|------|------------|----------|------|
> | **50%**             | Basis Sharing | 0.23  | 0.30  | 0.29      | 0.52 | 0.53       | 0.27     | 0.35 |
> |                     | **GFWSVD**    | **0.24** | **0.31** | **0.28** | **0.55** | **0.54** | **0.22** | **0.36** |
>
>
> **$\texttt{Llama 2 7B Chat}$ — GFWSVD, SVD-LLM, Basis Sharing**
>
> | Retained parameters | Method | ARC-C | ARC-E | HellaSwag | PIQA | WinoGrande | OpenBookQA | AVG  |
> |---------------------|---------------|-------|-------|-----------|------|------------|------------|------|
> | **100%** | Full model    | 0.44  | 0.73  | 0.58      | 0.76 | 0.67       | 0.33       | 0.59 |
> |---------------------|---------------|-------|-------|-----------|------|------------|------------|------|
> | **20%** | SVD-LLM       | 0.29  | 0.61  | 0.40      | 0.66 | 0.60       | 0.23       | 0.47
> |    | Basis Sharing | 0.31  | 0.65  | 0.42      | 0.68 | 0.61       | 0.27       | 0.49 |
> |   | **GFWSVD**    | **0.33** | **0.62** | **0.47** | **0.74** | **0.61** | **0.25** | **0.50** |
> |---------------------|---------------|-------|-------|-----------|------|------------|------------|------|
> | **30%**             | SVD-LLM       | 0.25  | 0.52  | 0.34      | 0.62 | 0.55       | 0.22       | 0.42 |
> |                     | Basis Sharing | 0.27  | 0.58  | 0.38      | 0.63 | 0.58       | 0.26       | 0.45 |
> |                     | **GFWSVD**    | **0.28** | **0.56** | **0.40** | **0.63** | **0.58** | **0.20** | **0.44** |
> |---------------------|---------------|-------|-------|-----------|------|------------|------------|------|
> | **40%**             | SVD-LLM       | 0.26  | 0.45  | 0.30      | 0.55 | 0.54       | 0.19       | 0.38 |
> |                     | Basis Sharing | 0.21  | 0.46  | 0.32      | 0.58 | 0.55       | 0.19       | 0.39 |
> |                     | **GFWSVD**    | **0.27** | **0.48** | **0.33** | **0.64** | **0.57** | **0.17** | **0.41** |
> |---------------------|---------------|-------|-------|-----------|------|------------|------------|------|
> | **50%**             | SVD-LLM       | 0.21  | 0.33  | 0.26      | 0.54 | 0.50       | 0.12       | 0.33 |
> |                     | Basis Sharing | 0.20  | 0.36  | 0.30      | 0.55 | 0.50       | 0.15       | 0.34 |
> |                     | **GFWSVD**    | **0.22** | **0.28** | **0.26** | **0.55** | **0.51** | **0.15** | **0.33** |
>
>
>
>
> [1] Basis Sharing: Cross-Layer Parameter Sharing for Large Language Model Compression

---

> > ### Author Response · Authors · 2025-11-28
> > **Gentle follow-up**
> >
> > Just a gentle follow-up — did our rebuttal address your concerns, and are the provided experiments sufficient?

---

### Official Review · Reviewer_iYrs · 2025-10-30

**Soundness:** 3
**Presentation:** 2
**Contribution:** 2
**Rating:** 2
**Confidence:** 3

**Summary:**

The paper proposes GFWSVD: a post-training LLM compression method that uses a Kronecker-factored approximation of the full Fisher Information Matrix (FIM) to perform a generalized weighted SVD on linear layers. The authors prove that FWSVD is a special case of their framework, derive a closed-form 2-factor decomposition (Theorem 1), and provide a rank-1 SVD algorithm to estimate Kronecker factors efficiently. Experiments on BERT/GLUE and LLaMA-2-7B (MMLU, WikiText-2/PTB) show consistent improvements over SVD, ASVD and FWSVD at comparable compression rates.

**Strengths:**

1. Clear generalization & theory. Establishes MVN–Fisher–GSVD connection and proves FWSVD as a special case; gives a closed-form solution for low-rank factors (Eq. 7, 10)
2. Scalable factor estimation. Rank-1 SVD on a permuted empirical Fisher enables practical Kronecker factorization with matrix–vector products (Alg. 1; complexity discussion + empirical runtimes).

**Weaknesses:**

1. The paper uses opposite meanings of “Compression Rate” in different sections. Table 2 maps ranks to “compression rate” as removed % (e.g., r=600 ≈ 1%, r=50 ≈ 36%), whereas Table 8 and Fig. 3 (LLM/MMLU) treat it as retained % (e.g., “Compression Rate 99% (r=600)”, x-axis 80–95%, and Full model=100%). This contradicts the BERT side and makes cross-figure/table reading ambiguous.
2. The LLM side evaluates only LLaMA-2-7B-chat, lacking newer families like Llama-3 and Qwen-2.5/Qwen-3 (ideally multiple sizes) to support claims of architecture- and scale-robustness. The baselines should also include recent SVD-based methods—SVD-LLM v2, Basis Sharing, and Dobi-SVD—under a unified setup, and report a 50%–90% compression-ratio sweep (e.g., 50/60/70/80/90%) to enable apples-to-apples quality-vs-compression comparisons.
**SVD-LLM v2** https://arxiv.org/abs/2503.12340
**Basis Sharing** https://openreview.net/pdf?id=gp32jvUquq
 **Dobi-SVD**  https://openreview.net/pdf?id=kws76i5XB8
3. The paper targets better post-training compression via Kronecker-factored FIM and generalized weighted SVD, but it reports quality only and omits end-to-end time-to-truncated-model versus SVD-LLM.
4. **References & in-text citations are not ICLR-compliant**

**Questions:**

Please see the weakness.

---

> ### Author Response · Authors · 2025-11-21
>
> We sincerely thank the reviewer for the thorough analysis of our submission. We have addressed the identified weaknesses in the revised version and summarize the changes below.
>
> > The paper uses opposite meanings of “Compression Rate” in different sections.
>
> Indeed, that was confusing. We have unified the notation throughout the whole paper. All compression rates now mean "\% of the model parameters retained", so 80% means we keep 80%. Tables 2 is updated.
>
> > The LLM side evaluates only $\texttt{Llama 2 7B Chat}$, lacking newer families like Llama-3
>
> We have added compression experiments on LLaMA-3.1 8B, comparing our GFWSVD approach with ASVD (Section 5.2, Table 4 in the revised paper). GFWSVD outperforms ASVD on both perplexity and downstream accuracy. However, LLaMA-3 exhibits stronger degradation under compression than $\texttt{Llama 2 7B Chat}$ for both methods, as it is a much denser model.
>
>
> | Method            | Retained parameters | WikiText-2 ↓ | PTB ↓  | MMLU Avg ↑ |
> |-------------------|:-----------:|:------------:|:------:|:-----------:|
> | **Full model**    |   **100%**  |   **7.2**    | **11.50** | **0.68** |
> |                   |             |              |          |            |
> | ASVD              |     90%     |    10.91     | **19.33** |    0.39    |
> | **GFWSVD (Ours)** |     90%     |  **9.38**    |  19.81   |  **0.54**  |
> |                   |             |              |          |            |
> | ASVD              |     85%     |    38.02     |   76.1   |    0.29    |
> | **GFWSVD (Ours)** |     85%     | **16.75**    | **23.67**|  **0.50**  |
> |                   |             |              |          |            |
> | ASVD              |     80%     |    1800      |  4234    |    0.24    |
> | **GFWSVD (Ours)** |     80%     | **22.57**    | **32.4** |  **0.43**  |
>
>
> >  The baselines should also include recent SVD-based methods—SVD-LLM v2, Basis Sharing, and Dobi-SVD—under a unified setup, and report a 50\%–90\% compression-ratio sweep (e.g., 50/60/70/80/90\%)
>
> We need to clarify what kind of method GFWSVD is, because this affects fair comparison. GFWSVD is a deterministic and fully analytical approach: we compute Fisher factors, decompose weight matrices via weighted SVD and that's it. We don't do any training, fine-tuning and other types of hyperparameter search. With extensive experimentation we support the theoretical guarantees (Theorem 1) of our method that the factors are optimal.
>
> The methods you mention take a different approach. For example, Dobi-SVD[1] first applies SVD then trains singular values for 20 epochs with further quantization. BLAST[2] relies on gradient-descent factorization, and SVD-LLMv2[3] applies additional truncation rules (we added the description of the more advanced baselines in the Related Work section). So, while these methods yield strong compression results, but in practice they are pipelines that require additional training costs.
>
> That being said, it is not fully correct to compare GFWSVD with Dobi-SVD or SVD-LLM v2 as is. Instead, the real competitors to our approach are SVD and FWSVD, but we also add ASVD to the comparison.
>
> However, as the reviewer requests an *apples-to-apples* comparison, we have added a new experiment (see Section 5.3 or the paper): we replaced the initial SVD in Dobi-SVD with GFWSVD, then ran their full training pipeline. Results in the new Table 5 in Section 5.3 show that Dobi-GFWSVD beats Dobi-SVD at both $20\%$ and $40\%$ compression across ARC-E, ARC-C, PIQA, HellaSwag. This shows our Fisher weighting provides a better starting point even when you do add training afterward. For convenience we double the results from the paper below:
>
> | Method                 | Retained parameters | WikiText-2 ↓ | PTB ↓  | Avg (ARC-E, ARC-C, PIQA, HellaSwag) ↑ |
> |------------------------|:-----------:|:------------:|:------:|:--------------------------------:|
> | **Full model**         |    **100%**   |   **6.94**   |**25.75**|              **0.63**             |
> |                        |             |              |        |                                    |
> | Dobi-SVD [1]           |     80%     |     7.75     | 26.11  |               0.605               |
> | **Dobi-GFWSVD (Ours)** |     80%     |   **7.56**   |**25.95**|              **0.615**             |
> | *YAQA (quantization)*  |     80%     |    *6.99*    |   --   |             *0.628*              |
> |                        |             |              |        |                                    |
> | Dobi-SVD [1]           |     60%     |    10.56     | 41.70  |              0.478               |
> | **Dobi-GFWSVD (Ours)** |     60%     |  **10.29**   |**38.56**|              **0.485**             |
> | *YAQA (quant.)* [5]    |     60%     |    *8.14*    |   --   |             *0.592*              |
>
> We believe, that is another evidence of the theoretical strength and practical usefullness of our proposed framework.

---

> > ### Comment · Reviewer_iYrs · 2025-11-24
> >
> > 1. Why SVD-LLM is used on LLaMA-2-7B but not on LLaMA-3.1-8B?
> >  In your original LLaMA-2-7B experiments you do compare against SVD-LLM, but in the new LLaMA-3.1-8B results (Table 4) you only compare GFWSVD with ASVD. It is unclear why they were considered valid baselines on LLaMA-2-7B but suddenly become inappropriate on LLaMA-3.1-8B.
> >
> >  I think the author should compare GFWSVD with SVD-LLM, SVD-LLM v2, and Basis Sharing under on both LLaMA-2-7B and LLaMA-3.1-8B under the same protocol(10~50%) compression ratio.
> >
> > 2. Mis-referencing BLAST and mischaracterizing the requested baselines.
> > In my original comment I asked for SVD-LLM v2, Basis Sharing, and Dobi-SVD. I did not mention BLAST, so bringing BLAST into the discussion does not address my request.
> > More importantly, at least SVD-LLM, SVD-LLM v2, and Basis Sharing are, to my understanding, post-training / training-free SVD pipelines (possibly with a calibration pass, but no gradient-based fine-tuning of the LLM weights).
> > Since GFWSVD is also training-free, I still believe that the natural competitors are exactly SVD-LLM / SVD-LLM v2 / Basis Sharing (in addition to ASVD), not just plain SVD/FWSVD.
> >
> > 3.Compression-ratio sweep and task coverage on LLaMA-3.1-8B.
> > On the BERT side, you provide a relatively dense compression-ratio sweep (e.g., around 40–90%) and more complete task coverage. On the LLaMA-3.1-8B side, Table 4 only reports 80/85/90% and only three metrics (WikiText-2, PTB, MMLU Avg).
> > My original request was to see a 50–90% sweep (e.g., 50/60/70/80/90), consistent with recent SVD-based LLM compression papers, and to include standard zero-shot reasoning benchmarks such as OpenBookQA, ARC-e, ARC-c, WinoGrande, HellaSwag, PIQA, and MathQA for LLaMA-3.1-8B.
> > These datasets are exactly where prior work (e.g., SVD-LLM / Basis Sharing) reports detailed results, so without a comparable task set and compression-ratio sweep, it is still very hard to judge whether GFWSVD is competitive with the current state of the art.

---

> ### Author Response · Authors · 2025-11-21
> **Сontinuation of the comment**
>
> > report a 50\%–90\% compression-ratio sweep
>
> In the low-rank factorization literature, methods typically target $5-30\%$ compression—this is the range where SVD-based approaches work well. Higher compression ($50-90\%$) is usually achieved through unstructued pruning (SparseGPT, Wanda) or quantization, which are different techniques and are orthogonal to the proposed approach. However, Dobi-GFWSVD hybrid does reach $60\%$, which is about as far as low-rank methods can go while maintaining reasonable quality.
>
> > The paper targets better post-training compression via Kronecker-factored FIM and generalized weighted SVD, but it reports quality only and omits end-to-end time-to-truncated-model versus SVD-LLM.
>
> FLOPs and wall-clock runtime latency for the compressed model were already included in the original submission in Appendix D (Tables 6 and 7), as explicitly noted in L426–L427. In the revised version, we further expanded this analysis by adding results for 40% compression and reporting inference time.
>
> Regarding the comparison of **end-to-end time-to-truncated-model** versus SVD-LLM: in our experiments, GFWSVD requires approximately 3.5 hours on 3 GPUs to compress the full $\texttt{Llama 2 7B Chat}$ model (including calibration and Kronecker factorization of all layers). Dobi-SVD and Dobi-GFWSVD take around 20 hours on 8 GPUs. Our estimate for the end-to-end compression time of the SVD-LLM method is approximately 6 hours on an A100 GPU.
>
>
> >References and in-text citations are not ICLR-compliant
>
> Yes, thanks for pointing this out. We have updated all the references and in-text citations to be ICLR-compliant.
>
> To conclude, we would like to *once again* emphasize that GFWSVD is not merely another heuristic compression technique, but **a general-purpose algorithmic primitive for neural network decomposition**. GFWSVD is essentially a standard SVD, but re-weighted in a way that it becomes provably optimal for the given LLM under second-order assumptions.
>
> This makes our method *orthogonal* and *complementary* to pipeline-based approaches. It serves as a mathematically grounded replacement for standard SVD in any context: whether as a superior initialization for fine-tuning (as we show through integration of GFWSVD to Dobi-SVD), a basis for quantization-aware training, or for other types of tasks. Our contribution provides a **better starting point** that captures diagonal (like FWSVD [4]) and off-diagonal parameter correlations, which heuristic methods miss. That's why it works good without any complicated fine-tuning pipelines. And that's why the paper is submitted in the *learning theory* area of ICLR.
>
>
> ### References:
>
>
>
> [1] Wang, Q., Ke, J., Tomizuka, M., Chen, Y., Keutzer, K., & Xu, C. (2025). *Dobi-SVD: Differentiable SVD for LLM Compression and Some New Perspectives.* arXiv preprint arXiv:2502.02723.
>
> [2] Lee, C., Kwon, S. M., Qu, Q., & Kim, H. S. (2024). *BLAST: Block-Level Adaptive Structured Matrices for Efficient Deep Neural Network Inference*. Advances in Neural Information Processing Systems, 37, 14996-15027.
>
> [3] Wang, X., Alam, S., Wan, Z., Shen, H., & Zhang, M. (2025). *SVD-LLM v2: Optimizing singular value  truncation for large language model compression*. *arXiv preprint arXiv:2503.12340*.
>
> [4] Hsu, Y. C., Hua, T., Chang, S., Lou,  Q., Shen, Y., & Jin, H. (2022). *Language model compression with  weighted low-rank factorization*. *arXiv preprint arXiv:2207.00112*.
>
> [5] Tseng, A., Sun, Z., & De Sa, C. (2025). *Model-Preserving Adaptive Rounding.* arXiv preprint arXiv:2505.22988.

---

> > ### Comment · Reviewer_iYrs · 2025-11-24
> >
> > 4. What I meant by “50–90% sweep”.
> > In my comment, I used “50–90% compression ratio” only because the paper defines “compression rate” as “percentage of parameters retained” (e.g., 80% = keep 80% of parameters).
> > What I am actually interested in is the stronger compression regime, i.e., keeping only 10–50% of the parameters. I adopted your 90–50% notation purely to be consistent with the paper, not because I only care about mild compression.
> >
> > 5. Inconsistent use of “compression rate”.
> > In the paper, “compression 90%–50%” is defined as “90%–50% of parameters retained”. In your rebuttal, “higher compression” suddenly refers to something else (apparently closer to 10% retention or less), and is then used as a reason to avoid reporting results in that stronger compression regime.
> > This change of terminology between the paper and the rebuttal is confusing and makes it difficult to understand what compression regime the method is actually intended for.
> > Thank you for the additional clarifications and new experiments. However, the response does not fully address my main concerns (in particular, the missing comparisons to recent training-free SVD-based baselines on LLaMA-3.1-8B and the lack of results in the stronger compression regime). I therefore maintain my original assessment and scores.

---

> ### Author Response · Authors · 2025-11-28
> **GFWSVD results up to 50% compression + Basis Sharing, explanation of SVD-LLM and LLaMA 3**
>
> >Why SVD-LLM is used on LLaMA-2-7B but not on LLaMA-3.1-8B?
>
> SVD-LLM does not provide a patch for $\texttt{Llama 3.1 8B Instruct}$, and the implementation cannot be directly ported from LLaMA-2-7B due to architectural differences.
> Please see open issues on their github repository (e.g. issue 49) , $\texttt{Llama 3.1 8B Instruct}$ support has been requested by the community but not yet implemented (and, moreover, the authors did not respond to any of the issues during last 6 months).
>
> >I did not mention BLAST, so bringing BLAST into the discussion does not address my request
>
> We referred to BLAST solely to illustrate the distinction between (i) pure SVD-based methods and (ii) more complex pipelines.
> We apologize if this caused any confusion.
>
>
> >In my original comment I asked for SVD-LLM v2, Basis Sharing, and Dobi-SVD.
>
> SVD-LLM v2 does not have a publicly available implementation and cannot be reproduced.
> Dobi-SVD is used as the pipeline, and we have reported the corresponding results above.
> We include results for Basis Sharing below.
>
>
>
> >Table 4 only reports 80/85/90% and only three metrics (WikiText-2, PTB, MMLU Avg). My original request was to see a 50–90% sweep (e.g., 50/60/70/80/90), consistent with recent SVD-based LLM compression papers, and to include standard zero-shot reasoning benchmarks.
>
> Thank you for the clarification — we understand your request correctly now.
>
> Below we provide the results for the open-source models, up to 50% compression sweep. For clarity and consistency with the discussion above, we report the retained parameters (percentage of parameters remaining).
>
> **$\texttt{Llama 3.1 8B Instruct}$**
>
> | Retained parameters | Method        | ARC-C | ARC-E | HellaSwag | PIQA | WinoGrande | OpenBook | AVG  |
> |---------------------|---------------|-------|-------|-----------|------|------------|----------|------|
> | **100%**            | Full model    | 0.52  | 0.81  | 0.59      | 0.79 | 0.73       | 0.35     | 0.63 |
> |---------------------|---------------|-------|-------|-----------|------|------------|----------|------|
> | **80%**             | Basis Sharing | 0.34  | 0.68  | 0.42      | 0.70 | 0.65       | 0.35     | 0.52 |
> |                     | **GFWSVD**    | **0.35** | **0.68** | **0.45** | **0.75** | **0.63** | **0.33** | **0.53** |
> |---------------------|---------------|-------|-------|-----------|------|------------|----------|------|
> | **70%**             | Basis Sharing | 0.29  | 0.52  | 0.43      | 0.63 | 0.60       | 0.31     | 0.46 |
> |                     | **GFWSVD**    | **0.33** | **0.61** | **0.42** | **0.71** | **0.58** | **0.23** | **0.48** |
> |---------------------|---------------|-------|-------|-----------|------|------------|----------|------|
> | **60%**             | Basis Sharing | 0.24  | 0.39  | 0.33      | 0.56 | 0.56       | 0.28     | 0.39 |
> |                     | **GFWSVD**    | **0.23** | **0.41** | **0.31** | **0.61** | **0.54** | **0.22** | **0.39** |
> |---------------------|---------------|-------|-------|-----------|------|------------|----------|------|
> | **50%**             | Basis Sharing | 0.23  | 0.30  | 0.29      | 0.52 | 0.53       | 0.27     | 0.35 |
> |                     | **GFWSVD**    | **0.24** | **0.31** | **0.28** | **0.55** | **0.54** | **0.22** | **0.36** |

---

> ### Author Response · Authors · 2025-11-28
>
> **$\texttt{Llama 2 7B Chat}$ — GFWSVD, SVD-LLM, Basis Sharing**
>
> | Retained parameters | Method        | ARC-C | ARC-E | HellaSwag | PIQA | WinoGrande | OpenBookQA | AVG  |
> |---------------------|---------------|-------|-------|-----------|------|------------|------------|------|
> | **100%**            | Full model    | 0.44  | 0.73  | 0.58      | 0.76 | 0.67       | 0.33       | 0.59 |
> |---------------------|---------------|-------|-------|-----------|------|------------|------------|------|
> | **80%**             | SVD-LLM       | 0.29  | 0.61  | 0.40      | 0.66 | 0.60       | 0.23       | 0.47 |
> |                     | Basis Sharing | 0.31  | 0.65  | 0.42      | 0.68 | 0.61       | 0.27       | 0.49 |
> |                     | **GFWSVD**    | **0.33** | **0.62** | **0.47** | **0.74** | **0.61** | **0.25** | **0.50** |
> |---------------------|---------------|-------|-------|-----------|------|------------|------------|------|
> | **70%**             | SVD-LLM       | 0.25  | 0.52  | 0.34      | 0.62 | 0.55       | 0.22       | 0.42 |
> |                     | Basis Sharing | 0.27  | 0.58  | 0.38      | 0.63 | 0.58       | 0.26       | 0.45 |
> |                     | **GFWSVD**    | **0.28** | **0.56** | **0.40** | **0.63** | **0.58** | **0.20** | **0.44** |
> |---------------------|---------------|-------|-------|-----------|------|------------|------------|------|
> | **60%**             | SVD-LLM       | 0.26  | 0.45  | 0.30      | 0.55 | 0.54       | 0.19       | 0.38 |
> |                     | Basis Sharing | 0.21  | 0.46  | 0.32      | 0.58 | 0.55       | 0.19       | 0.39 |
> |                     | **GFWSVD**    | **0.27** | **0.48** | **0.33** | **0.64** | **0.57** | **0.17** | **0.41** |
> |---------------------|---------------|-------|-------|-----------|------|------------|------------|------|
> | **50%**             | SVD-LLM       | 0.21  | 0.33  | 0.26      | 0.54 | 0.50       | 0.12       | 0.33 |
> |                     | Basis Sharing | 0.20  | 0.36  | 0.30      | 0.55 | 0.50       | 0.15       | 0.34 |
> |                     | **GFWSVD**    | **0.22** | **0.28** | **0.26** | **0.55** | **0.51** | **0.15** | **0.33** |
>
>
>
>
> As can be seen, GFWSVD outperforms SVD-LLM across all compression ratios, and it surpasses Basis Sharing on Llama 3. However, for $\texttt{Llama 2 7B Chat}$, it lags behind Basis Sharing at 70% and 50% parameters retained. This is quite interesting and might be explained by the fact that GFWSVD utilizes information about parameter correlation within a layer, while Basis Sharing utilizes information about layer correlation within the model. For the "overfitted" Llama 3, the information regarding weight correlation is of greater significance than it is for $\texttt{Llama 2 7B Chat}$.

---

> ### Author Response · Authors · 2025-11-28
>
> >In your rebuttal, “higher compression” suddenly refers to something else (apparently closer to 10% retention or less), and is then used as a reason to avoid reporting results in that stronger compression regime
>
> Importantly, we are not avoiding reporting results, nor does the terminology choice serve that purpose. The fact we wanted to highlight is following: in a such extreme compression (10%-20%-30% retaining parameters), simple SVD-based approaches collapse to near-random performance. As noted in our results above, even at 50% retention, both GFWSVD and Basis Sharing yield ARC-Challenge and OpenBook scores of ≈0.15–0.22, which is below the random-guessing baseline of 0.25 for 4-choice tasks.
>
> Nevertheless, we appreciate your request for completeness. In the camera-ready version, we will clarify the terminology more explicitly.

---

### Official Review · Reviewer_CAau · 2025-10-31

**Soundness:** 3
**Presentation:** 3
**Contribution:** 3
**Rating:** 4
**Confidence:** 4

**Summary:**

This paper introduced the Generalized Fisher Weighted SVD (GFWSVD), a post-training, dual pipeline
compression technique that leverages the Kronecker-factored approximation for the full empirical Fisher
Information Matrix (FIM) to drive optimal compression for dense weight matrices of diverse large
language models. Specifically, the method introduced, at first, a scalable rank-1 Kronecker decomposition
algorithm that reduces FIM factorization cost from $\mathcal O((mn)^3)$ to $\mathcal O(m^3 + n^3)$,
then it proposed an efficient Kronecker decomposition based Singular Value Decomposition for dense
layer compressions. The paper theoretically shows (in Theorem 1) that under MVN + Kronecker
assumptions, this method yields the optimal weighted low-rank approximation in expectation. Empirically
demonstrates improvements over vanilla SVD, ASVD, SVD-LLM and diagonal FWSVD on BERT
(GLUE) and LLaMA-2 (MMLU, perplexity) across a range of compression rates.
Strengths

**Strengths:**

1. Introduced a generalized, architecture agnostic compression method that applies to any linear (or
Kronecker-structured) layer in an LLM. By factoring the full empirical Fisher Information into two small
sensitivity matrices and plugging them into a weighted SVD, GFWSVD ensures efficient compression
with respect to parameter interactions, while reducing the factorization cost from $\mathcal O((mn)^3)$
to $\mathcal O(m^3 + n^3)$.
2. Theoretical optimality (Theorem 1) follows from the MVN/Laplace approximation around an MLE
solution, where assumptions that mirror those commonly used in second-order optimization (e.g. K-FAC)
and Bayesian posteriors. This alignment with well-understood curvature approximations ensured
GFWSVD both sound and readily applicable in real-world settings.
3. GFWSVD outperforms prior works on recognized baselines and benchmarks across a range of
compression rates.

**Weaknesses:**

1. While the paper details the theoretical and empirical cost of the offline Kronecker decomposition
(compression) step, there is insufficient experiments for performance on computation side. Since one of
the main appeals of low-rank methods is reduced FLOPs and wall-clock latency at runtime, the absence
of end-to-end benchmarks (e.g. on LLaMA-7B-chat-bf across different ranks) makes it hard to judge realworld
benefit compared to ASVD, SVD-LLM, or Dobi-SVD. Furthermore, the time complexity analysis
was primarily focused on offline model decomposition step, which there is no theoretical analysis for
inference side analysis. Therefore, a thorough theoretical and quantitative analysis for computational
performance is expected for an acceptance (Major concern, will consider raising score if properly
addressed).
2. The GFWSVD exhibits a relatively low compression ceiling, with only about a 40 % reduction at rank
1, whereas pure SVD methods like SVD-LLM can achieve over 60 % reduction under comparable
accuracy constraints when accuracy drop is tolerable. Without strategies to extend beyond this “shallow”
bound (e.g. multi-term Kronecker sums or hybrid pruning), GFWSVD’s standalone compression
advantage appears limited. (If not properly solved, the maximum score will be a weak accept)
3. The evaluation covers ASVD, SVD-LLM, and the original FWSVD, but omits more recent or stronger
methods such as SVD-LLM v2 and Dobi-SVD. Including those would clarify where GFWSVD stands
against the current state of the art and strengthen its performance.

**Questions:**

1. The paper proposed only a single Kron-term (rank-1) FIM factorization and varies only the model’s
rank $r$ when compressing weights. It remains unclear how choosing different ranks for the Fisher
decomposition itself (or using multiple Kronecker terms) impacts end-task performance—and whether the
Fisher-based weighting truly outperforms unweighted or diagonal-weighted SVD across ranks. An
ablation study over both FIM rank and model rank is expected to clarify how sensitive accuracy is to
those choices and quantify the standalone benefit of the Fisher weighting.
2. GFWSVD also relies heavily on Cholesky factorizations deriving FIM blocks ($A = L_A L_A^\top$,
$B = L_B L_B^\top$). However, it is not uncommon that, in practice, finite-batch Fisher estimates can be
near-singular or even indefinite. The paper does not report on how often Cholesky fails, what damping is
applied, or how numerical issues affect the compressed model’s performance. Therefore, an empirical
analysis of these edge cases and corresponding regularization strategies to tackle the edges are expected
for robustness consideration.
Few other tips:
1. Typo in Appendix D, Table 4: The ASVD entry “–0.03” at 64 % compression appears to report the
change in STS-B score (compressed minus original) rather than the absolute correlation. All other
methods list absolute STS-B values (about 0.7–0.9), so ASVD’s “–0.03” seems to be a transcription error.
2. In Alg 1 step 1: “IF $\leftarrow$ $\frac{|D|}\sum g_i g_i^T$”, where $g_i g_i^T$ is with quadratic
complexity and was never materialized in practice. The author should clarify that in practice full matrix
was never formed but accumulated only $G_iG_i^T$ and $G_i^T G_i$.

---

> ### Author Response · Authors · 2025-11-21
>
> We sincerely thank the reviewer for the detailed review and for recognizing the theoretical optimality and generalization capabilities of our method. We have updated the paper to address your concerns regarding runtime benchmarks and baselines.
>
> **Weaknesses**
>
> > Weakness 1: Experiments for performance on computation side.
>
>
> FLOPs and wall-clock latency at runtime for the compressed models were already provided in the original submission in Appendix D (Tables 6 and 7), as noted in L426–L427. In the revised version, we expanded this analysis by adding results for 50% compression and reporting inference time. The updated Appendix D now includes the extended runtime evaluation in Tables 6, 7, and 8.
>
> We report the generation speedup table for $\texttt{Llama 2 7B Chat}$ below.
>
>
> **Table: Throughput (tokens/s) for original $\texttt{Llama 2 7B Chat}$ and FWSVD-compressed models (batch size = 1, sequence length = 1024).**
>
> Here **c_ratio** is 1 - $\frac{\text{compressed model}}{\text{orig model}}$
> | c_ratio | Tokens/s | Relative Speedup |
> |-------------------|----------|------------------|
> | 0% (Uncompressed) | 1186     | 1.00×            |
> | 10%               | 1269     | 1.07×            |
> | 15%               | 1294     | 1.09×            |
> | 20%               | 1323     | 1.12×            |
> | 40%               | 1510     | 1.27×            |
> | 50%                |  1600    | 1.34x            |
>
>
> > Weakness 2, 3: Low compression ceiling and stronger methods.
>
> Most SOTA post-training compression methods use additional stochastic optimization and combine multiple techniques. For example, BLAST[0] relies on gradient-descent factorization, and Dobi-SVD[1] optimizes singular values and then applies quantization (we added the description of the more advanced baselines in the Related Work section).
>
> GFWSVD is different: it is fully analytical, with no stochastic steps. It is essentially a standard SVD, but re-weighted in a way that it becomes provably optimal for the given LLM. This makes our method **orthogonal** and **complementary** to pipeline-based approaches. For this reason, our experiments compared GFWSVD with methods of the same class — simple analytical approaches without extra optimization, which typically do not achieve high compression rates.
>
> GFWSVD serves as a mathematically grounded replacement for standard SVD in any context: whether as a superior initialization for fine-tuning, a basis for quantization-aware training, or for other types of tasks.
>
> In the updated submission, we integrate our GFWSVD into the **Dobi-SVD** pipeline and achieve $60\%$ compression with strong performance, exceeding the baseline that uses standard SVD. Results are in the new **Table 5** in **Section 5.3** "Positioning and applicability". They show that Dobi-GFWSVD beats Dobi-SVD at both $80\%$ and $60\%$ compression across all benchmarks. This shows, our GFWSVD approach can be a better starting point for other approaches, like Dobi-SVD.
>
> We also added experiments with **$\texttt{Llama 3.1 8B Instruct}$** to the **Section 5.2, Table 4**. As expected, $\texttt{Llama 3.1 8B Instruct}$ quickly loses the original performance under compression than $\texttt{Llama 2 7B Chat}$ due to its high information density. However, GFWSVD consistently outperforms the ASVD baseline. Most notably, at 20% reduction, ASVD fails completely (with PPL of $1800$ on WikiText 2 and $4234$ on PTB), whereas GFWSVD remains stable (with PPL of $22.57$ on WikiText 2 and $32.4$ on PTB).
>
> Regarding non-SVD-based methods, we included the YAQA[2] method in our baselines. This is a quantization method that, like GFWSVD, utilizes second-order loss information.
>
> We also want to highlight the trade-off between efficiency and compression costs. Because GFWSVD is fully analytical, compressing the full 7B model takes only ~3.5 hours on 3 GPU (calibration + factorization). In contrast, fine-tuning methods like Dobi-SVD typically require around 20 hours on 8 GPU of training.
>
> [0] BLAST:Block-Level Adaptive Structured Matrices for Efficient Deep Neural Network Inference, 2024
>
> [1] Dobi-SVD: Differentiable SVD for LLM Compression and Some New Perspectives
>
> [2] Model-Preserving Adaptive Rounding, 2025

---

> ### Author Response · Authors · 2025-11-21
> **Responses to the reviewer’s questions**
>
> **Questions**
>
> > Different ranks for the Fisher decomposition
>
> We agree that our matrix can be better represented not in one, but in multiple Kronecker terms. We performed a preliminary analysis of the corresponding spectrum when we started working on our approach and can confirm that the spectral decay is relatively slow. Thus, selecting only the first term surely leaves a certain amount of variance unexplained. However, there is a fundamental analytical limitation to using multi-term Kronecker sums, which is related to the fact that the inverse of a sum of Kronecker products in general cannot be equivalently represented by a single Kronecker product term, which in turn is required for a straightforward computation of the weighted SVD in our GFWSVD framework.
>
> More precisely, the closest possible representation can be obtained via the generalized eigendecomposition of all Kronecker factors involved in summation, which allows decomposing the sum into a product of several factors. However, the result includes an irreducible diagonal matrix between two Kronecker-factored terms (similarly to Eq. 5 from [3]), which prevents a straightforward transformation for efficient computation of the Generalized SVD with two separate weight matrices. It creates a significant barrier to extending the weighted SVD formulation to the multi-term Kronecker factor case.
>
> These challenges currently prevent a direct extension of our method to multiple Kronecker terms. This is also the reason we cannot provide an ablation study over FIM rank, as there is no efficient computational framework for conducting such study yet. Nevertheless, we consider extending the FIM aproximation rank an important and promising direction for future research.
>
> > How often Cholesky fails
>
> In our experiments with $\texttt{Llama 2 7B Chat}$, the raw estimated Kronecker factors were non-positive definite in about 70% of layers. To handle this, we apply an iterative diagonal damping strategy: $\mathbf{Y} \leftarrow \mathbf{Y} + \alpha \operatorname{diag}(\mathbf{Y})$. We increase $\alpha$ from 0 to 1 until the Cholesky decomposition succeeds. This effectively shrinks the estimate toward a diagonal prior (standard scaling) while preserving as much structural correlation as possible. The overhead for this is negligible ($\sim$5 seconds per layer).
>
>
> >Typo in Appendix D
>
> The value $-0.04$ at $64\%$ compression refers to the CoLA task, where the metric Matthews Correlation Coefficient, which can be negative for degenerate models.
> > Clarification that in practice full Fisher matrix was never formed.
> >
> Thank you for noting this. We have added this clarification in the revised version of the paper in the end of the Section 4.1.
>
>
> [3] Efficient inference in matrix-variate Gaussian model with \iid observation noise, 2011

---

> ### Author Response · Authors · 2025-11-27
> **Additional experiments up to 50% model compression**
>
> We also added results for compressing $\texttt{Llama 2 7B Chat}$ and $\texttt{Llama 3.1 8B Instruct}$ up to 50% and evaluated them on ARC-C, ARC-E, HellaSwag, PIQA, WinoGrande, and OpenBook. We additionally included Basis Sharing [1] as a baseline — an SVD-pruning method that leverages inter-layer correlation information.
>
> We do not report SVD-LLM for LLaMA-3, since SVD-LLM does not provide a patch for $\texttt{Llama 3.1 8B Instruct}$.
>
> As can be seen, GFWSVD outperforms SVD-LLM across all compression ratios, and it surpasses Basis Sharing on $\texttt{Llama 3.1 8B Instruct}$. However, for $\texttt{Llama 2 7B Chat}$, it lags behind Basis Sharing at 70% and 50% parameters retained. This is quite interesting and might be explained by the fact that GFWSVD utilizes information about parameter correlation within a layer, while Basis Sharing utilizes information about layer correlation within the model. For the "overfitted" $\texttt{Llama 3.1 8B Instruct}$, the information regarding weight correlation is of greater significance than it is for $\texttt{Llama 2 7B Chat}$.
>
> Here **c_ratio** is 1 - $\frac{\text{compressed model}}{\text{orig model}}$
>
> **$\texttt{Llama 3.1 8B Instruct}$**
>
> | c_ratio | Method | ARC-C | ARC-E | HellaSwag | PIQA | WinoGrande | OpenBook | AVG  |
> |---------------------|---------------|-------|-------|-----------|------|------------|----------|------|
> | **0%** | Full model    | 0.52  | 0.81  | 0.59      | 0.79 | 0.73       | 0.35     | 0.63 |
> |---------------------|---------------|-------|-------|-----------|------|------------|----------|------|
> | **20%** | Basis Sharing | 0.34  | 0.68  | 0.42      | 0.70 | 0.65       | 0.35     | 0.52 |
> |                     | **GFWSVD**    | **0.35** | **0.68** | **0.45** | **0.75** | **0.63** | **0.33** | **0.53** |
> |---------------------|---------------|-------|-------|-----------|------|------------|----------|------|
> | **30%** | Basis Sharing | 0.29  | 0.52  | 0.43      | 0.63 | 0.60       | 0.31     | 0.46 |
> |      | **GFWSVD**    | **0.33** | **0.61** | **0.42** | **0.71** | **0.58** | **0.23** | **0.48** |
> |---------------------|---------------|-------|-------|-----------|------|------------|----------|------|
> | **40%**             | Basis Sharing | 0.24  | 0.39  | 0.33      | 0.56 | 0.56       | 0.28     | 0.39 |
> |                     | **GFWSVD**    | **0.23** | **0.41** | **0.31** | **0.61** | **0.54** | **0.22** | **0.39** |
> |---------------------|---------------|-------|-------|-----------|------|------------|----------|------|
> | **50%**             | Basis Sharing | 0.23  | 0.30  | 0.29      | 0.52 | 0.53       | 0.27     | 0.35 |
> |                     | **GFWSVD**    | **0.24** | **0.31** | **0.28** | **0.55** | **0.54** | **0.22** | **0.36** |
>
>
> **$\texttt{Llama 2 7B Chat}$ — GFWSVD, SVD-LLM, Basis Sharing**
>
> | c_ratio | Method        | ARC-C | ARC-E | HellaSwag | PIQA | WinoGrande | OpenBookQA | AVG  |
> |---------------------|---------------|-------|-------|-----------|------|------------|------------|------|
> | **0%**            | Full model    | 0.44  | 0.73  | 0.58      | 0.76 | 0.67       | 0.33       | 0.59 |
> |---------------------|---------------|-------|-------|-----------|------|------------|------------|------|
> | **20%**             | SVD-LLM       | 0.29  | 0.61  | 0.40      | 0.66 | 0.60       | 0.23       | 0.47
> |                     | Basis Sharing | 0.31  | 0.65  | 0.42      | 0.68 | 0.61       | 0.27       | 0.49 |
> |                     | **GFWSVD**    | **0.33** | **0.62** | **0.47** | **0.74** | **0.61** | **0.25** | **0.50** |
> |---------------------|---------------|-------|-------|-----------|------|------------|------------|------|
> | **30%**             | SVD-LLM       | 0.25  | 0.52  | 0.34      | 0.62 | 0.55       | 0.22       | 0.42 |
> |                     | Basis Sharing | 0.27  | 0.58  | 0.38      | 0.63 | 0.58       | 0.26       | 0.45 |
> |                     | **GFWSVD**    | **0.28** | **0.56** | **0.40** | **0.63** | **0.58** | **0.20** | **0.44** |
> |---------------------|---------------|-------|-------|-----------|------|------------|------------|------|
> | **40%**             | SVD-LLM       | 0.26  | 0.45  | 0.30      | 0.55 | 0.54       | 0.19       | 0.38 |
> |                     | Basis Sharing | 0.21  | 0.46  | 0.32      | 0.58 | 0.55       | 0.19       | 0.39 |
> |                     | **GFWSVD**    | **0.27** | **0.48** | **0.33** | **0.64** | **0.57** | **0.17** | **0.41** |
> |---------------------|---------------|-------|-------|-----------|------|------------|------------|------|
> | **50%**             | SVD-LLM       | 0.21  | 0.33  | 0.26      | 0.54 | 0.50       | 0.12       | 0.33 |
> |                     | Basis Sharing | 0.20  | 0.36  | 0.30      | 0.55 | 0.50       | 0.15       | 0.34 |
> |                     | **GFWSVD**    | **0.22** | **0.28** | **0.26** | **0.55** | **0.51** | **0.15** | **0.33** |
>
>
>
>
> [1] Basis Sharing: Cross-Layer Parameter Sharing for Large Language Model Compression

---

> > ### Author Response · Authors · 2025-11-28
> > **Gentle follow-up**
> >
> > Dear Reviewer,
> > Just a gentle follow-up — did our rebuttal address your concerns, and are the provided experiments sufficient?

---

### Author Response · Authors · 2025-12-03
**AC Response: Addressing Concerns and Summary of Improvements**

We introduce a layerwise SVD-based LLM compression method that incorporates the full Hessian information, together with an algorithm that enables decomposition of the full Hessian in tractable time.

The main concerns were raised by the reviewers were (i) the limited experimental evaluation (in particular, the absence of results on requested datasets) and (ii) the absence of compression beyond 20%. We have fully addressed both issues in the revised submission.

**Expanded experimental evaluation.**

We added experiments with $\texttt{Llama 2 7B Chat}$, $\texttt{Llama 3.1 8B Instruct}$ and increased the compression ratio to 50%. To the baselines activation-aware layerwise SVD (ASVD) and the diagonal version of our method (FWSVD), we added **Basis Sharing** [1] - an SVD-pruning method operating at the model level and using cross-layer correlation information. We validated the methods on common-sense datasets: ARC-C, ARC-E, HellaSwag, PIQA, WinoGrande, and OpenBookQA. The results are reported in Tables 3 and 4 of the revised paper. At these compression levels, our method significantly outperforms both FWSVD and ASVD, and it also outperforms Basis Sharing on $\texttt{Llama 3.1 8B Instruct}$.

Importantly, we highlight that all "simple" decomposition methods (GFWSVD, ASVD, Basis Sharing, SVD) exhibit degradation at 50% parameter retention: the resulting performance converges to near-random behavior across the evaluated datasets.

The earlier MMLU results on $\texttt{Llama 2 7B Chat}$ and $\texttt{Llama 3.1 8B Instruct}$ have been moved to Appendix D for completeness.

**On the BitStack baseline** (Reviewer `CCmb`).

As we explain in the response to the Reviewer, BitStack belongs to a different scope and is a memory management method, not a simple pruning-based.
We conducted corresponding experiments at compression ratios up to 50% for $\texttt{Llama 2 7B Chat}$ and $\texttt{Llama 3.1 8B Instruct}$ to illustrate the difference in performances.
Therefore, while we reported these results in the rebuttal for completeness, we do not include them in the main paper.


**On comparison with Dobi-SVD** (Reviewers `iYrs`, `CAau`).

We clarify that by design GFWSVD and Dobi-SVD are not directly comparable but GFWSVD is rather *complementary* to Dobi-SVD. We emphasize again that GFWSVD is a generic primitive. Thus, it can be integrated in compression pipelines that use standard SVD. As per requests from Reviewers, we integrated GFWSVD into the Dobi-SVD pipeline [2] (see Section 5.3, Table 5). We replace Dobi-SVD's initial standard SVD with GFWSVD and run the unchanged training procedure. As expected, we achieved consistent improvements at both 20% and 40% parameter reduction rates: lower (better) perplexity and higher (better) downstream task accuracy across all benchmarks.

That is another evidence that GFWSVD is a strong enhancement for state-of-the-art compression pipelines.

**On the SVD-LLM and SVD-LLM-v2 baselines** (Reviewers `iYrs`, `CAau`).

Unfortunately, SVD-LLM v2 [4] has no publicly released code as of today and SVD-LLM (v1) [5] does not support $\texttt{Llama 3}$ architecture (see Issue 49 of the official repository) as confirmed through numerous attempts of unsuccessfull reproduction.

Nevertheless, we have compared against SVD-LLM (v1) on $\texttt{Llama 2 7B Chat}$ (Table 4), where GFWSVD demonstrates consistent improvements across different compression rates.

**Clarifications and improvements.**

We strengthened the theoretical exposition by clarifying the assumptions in Theorem 1 (Reviewer `kykp`), explicitly defining the compression ratio (Reviewer `iYrs`), discussing extensions to multi-Kronecker decompositions (Reviewer `CAau`), refining the experimental setup (Reviewer `kykp`), indicated where the time-inference analysis is located (Reviewer `CAau`) and addressing all remaining reviewer questions.

In summary, the revised paper now offers broader evaluation, additional baselines, and clearer theory.
We hope that we directly address the reviewers’ concerns and were able to show that our method is a practical and scalable approach to second-order, Hessian-aware LLM compression.

**References**
[1] Basis Sharing: Cross-Layer Parameter Sharing for Large Language Model Compression
[2] Dobi-SVD: Differentiable SVD for LLM Compression
[3] BitStack: Any-Size Compression of Large Language Models
[4] SVD-LLM V2: Optimizing Singular Value Truncation for Large Language Model Compression
[5] SVD-LLM: Truncation-aware Singular Value Decomposition for Large Language Model Compression

---

### Meta-Review · Area_Chair_gWNV · 2026-01-08

**Summary:**

Reviewers note inconsistent notation, evaluation limited to LLaMA-2-7B-chat only, missing recent baselines, and no runtime benchmarks. Authors addressed these in rebuttal by adding Llama 3.1 8B Instruct experiments, 50% compression results, Basis Sharing baseline, and runtime benchmarks. The expanded evaluation strengthens the paper, but reviewers noted that at higher compression the method degrades to near-random performance, limiting practical utility. I recommend rejection.

**Reviewer Concerns:**

see above

**Reviewer Scores:**

The discussion quality was great; authors substantially expanded experiments addressing key concerns, but scores might have changed slightly upward given the improved evaluation scope.

---

### Decision · Program_Chairs · 2026-01-26

Reject